# Modulation of autoimmune pathogenesis by T cell-triggered inflammatory cell death

Katsuhiro Sasaki [1], Ai Himeno[1], Tomoko Nakagawa[1], Yoshiteru Sasaki[1], Hiroshi Kiyonari[2] & Kazuhiro Iwai [1]

T cell-mediated autoimmunity encompasses diverse immunopathological outcomes; however, the mechanisms underlying this diversity are largely unknown. Dysfunction of the tripartite linear ubiquitin chain assembly complex (LUBAC) is associated with distinct autonomous immune-related diseases. *Cpdm* mice lacking Sharpin, an accessory subunit of LUBAC, have innate immune cell-predominant dermatitis triggered by death of LUBAC-compromised keratinocytes. Here we show that specific gene ablation of *Sharpin* in mouse Treg causes phenotypes mimicking *cpdm*-like inflammation. Mechanistic analyses find that multiple types of programmed cell death triggered by TNF from tissue-oriented T cells initiate proinflammatory responses to implicate innate immune-mediated pathogenesis in this T cell-mediated inflammation. Moreover, additional disruption of the *Hoip* locus encoding the catalytic subunit of LUBAC converts *cpdm*-like dermatitis to T cell-predominant autoimmune lesions; however, innate immune-mediated pathogenesis still remains. These findings show that T cell-mediated killing and sequential autoinflammation are common and crucial for pathogenic diversity during T cell-mediated autoimmune responses.

[1] Department of Molecular and Cellular Physiology, Graduate School of Medicine, Kyoto University, Kyoto 606-8501, Japan. [2] Animal Resource Development Unit and Genetic Engineering Team, RIKEN Center for Life Science Technologies, 2-2-3 Minatojima Minami-machi, Chuou-ku, Kobe 650-0047, Japan. Correspondence and requests for materials should be addressed to K.I. (email: kiwai@mcp.med.kyoto-u.ac.jp)

The linear ubiquitin chain assembly complex (LUBAC) is a tripartite enzyme comprising Sharpin, HOIL-1L, and catalytic HOIP; its dysfunction is closely associated with autoinflammatory syndrome and immune deficiency in humans and mice[1–3]. LUBAC conjugates linear-type ubiquitin chains on target substrates, such as NF-κB essential modulator (NEMO) and receptor-interacting serine/threonine protein kinase 1 (RIPK1), to activate NF-κB signaling in response to exogenous triggers such as TNFα, IL-1β, and T cell receptor (TCR) stimulation[4–8]. Upon TNFα signaling, the ligase activity of LUBAC is requisite for protection against two types of programmed cell death, apoptosis and necroptosis; the latter is induced by generation of cytosolic death-inducible complex II comprising RIPK1, RIPK3, FAS-associated death domain protein (FADD), cellular FLICE (FADD-like IL-1β-converting enzyme)-inhibitory protein (cFLIP), and Caspase-8[9,10]. LUBAC subunits contribute differentially to the stability of the complex. Deletion of *Hoip* or *Hoil-1l* in mice, which eliminates the LUBAC complex, results in embryonic lethality due to aberrant TNFR1-mediated endothelial cell death and defective vascularization[10,11], whereas spontaneous mutant mice lacking *Sharpin* (called *cpdm* mice) are viable but develop chronic skin autoinflammation, which is triggered by death of keratinocytes[4,12–14].

In humans, autoinflammation is a self-directed immune disorder that manifests as chronic and recurrent inflammation. In most cases, it has a genetic etiology, leading to dysregulation of innate, but not adaptive, immune responses; this causes overproduction of proinflammatory cytokines such as IL-1β and TNFα, or exaggerated responsiveness to a steady-state level of stimulation by proinflammatory cytokines that may trigger release of other endogenous stimuli, including damage-associated molecular patterns (DAMPs), to aggravate innate immune-related inflammation[15,16]. Thus, autoinflammation is defined by various forms of myeloid cell-mediated systemic inflammation, without classical autoimmune characteristics such as high-titer autoantibodies or the presence of self-reactive T cells. Other studies suggest that *cpdm* mice manifest additional features. Studies of *cpdm Rag1*−/− mice and *cpdm*-derived bone marrow cell transfer experiments suggest that hematopoietic cells, including T cells and B cells, are dispensable for development of *cpdm* skin disease[17,18]. Furthermore, Sharpin-deficient skin transplanted onto nude mice develops autonomous inflammatory responses that clearly indicate that keratinocytes showing hypomorphic LUBAC expression are susceptible to autonomous cell death mediated by FADD-caspase-8-dependent apoptosis and RIPK1-RIPK3-mixed-lineage kinase domain-like protein (MLKL)-dependent necroptosis, resulting in autoinflammation even under steady-state conditions[19].

Nevertheless, recent studies imply the presence of autoimmune aspects in LUBAC hypomorphic disease: *cpdm* mice show impaired development and a reduced number of Foxp3+ regulatory T cells (Treg), a critical T cell subset for immunosuppression. In addition, adaptive transfer of Sharpin-sufficient Treg into neonatal *cpdm* mice alleviates inflammatory responses in various tissues, but does not improve dermatitis[20,21]. These reports imply that *cpdm* mice suffer from both autoimmune and autoinflammatory diseases, although they exhibit predominantly innate immune-mediated inflammation. Here, we examine the possibility that T cell-induced inflammation elicits an apparently innate immune-mediated pathogenesis, as observed in *cpdm* disease.

## Results

**Loss of Sharpin in Treg causes *cpdm*-like skin inflammation.** To examine the pathogenic potency provoked by loss of Sharpin

in Treg, we generated T$_{reg}$-specific Sharpin-deficient mice (*Sharpin^{fl/fl}Foxp3^{Cre}*). Expression of Sharpin by CD4+Foxp3+ Treg purified from the peripheral lymphoid tissues of *Sharpin^{fl/fl}Foxp3^{Cre}* mice was completely abolished, whereas that of HOIP fell, indicating a profound reduction in the amount of LUBAC complex (Fig. 1a, Supplementary Fig. 1A). However, *Sharpin^{fl/fl}Foxp3^{Cre}* mice exhibited few changes in the number and proportion of Foxp3+ thymocytes and peripheral Treg (Fig. 1b, Supplementary Fig. 1B). Partial impairment of the NF-κB signaling pathway in Sharpin-deficient Treg was demonstrated by a reduction in p65 phosphorylation on Ser536 and subtle inhibition of IκBα degradation during TCR stimulation; however, the TCR-mediated ERK signaling pathway was unaffected (Fig. 1c). The cell-intrinsic roles of Sharpin in T cells were confirmed in Sharpin-KO and HOIP-KO Jurkat or murine hybridoma cells. HOIP-KO Jurkat cells lost the ability to activate NF-κB signaling in response to TCR stimulation, whereas Sharpin-KO cells still retained this signaling pathway, albeit mildly impaired (Supplementary Fig. 1C, D). OVA agonist peptide (SIINFEKL)-driven secretion of IL-2 from a murine OVA-specific B3Z T cell hybridoma upon loss of either HOIP or Sharpin was attenuated, which indicated marked involvement of LUBAC subunits in TCR-mediated signaling (Supplementary Fig. 1E). Furthermore, introduction of HOIP mutants into HOIP-KO Jurkat cells revealed that the UBA domain, which is required for stable HOIP expression via interaction with the other LUBAC subunits (Sharpin and HOIL-1L) in various cells, was also critical in T cells. In addition, the novel zinc finger (NZF) domain of HOIP appeared essential due to its strong binding to polyubiquitin chains and/or NEMO. LUBAC ligase activity was dispensable for TCR-mediated NF-κB signaling since HOIP C885S (which lacks ligase activity) induced TCR- but not TNFα-mediated activation of NF-κB (Supplementary Fig. 1F, G)[22]. Thus, it is likely that the amount of LUBAC containing HOIP, but neither ligase activity nor composition of the complex, is the critical factor for TCR-mediated T cell activation (Supplementary Fig. 1G).

Treg from *Sharpin^{fl/fl}Foxp3^{Cre}* mice showed normal expression of T$_{reg}$ functional surface markers and stabilization markers for thymic Treg (Supplementary Fig. 1H, I), indicating that the trace amount of LUBAC (composed of HOIP and HOIL-1L) present in Sharpin-deficient cells is sufficient to elicit signaling for thymic T$_{reg}$ development despite mild impairment of TCR signaling. However, global gene expression between Sharpin-deleted Treg and control Treg was different (63 [<two-fold] and 84 [>two-fold] out of 21,178 detected genes). Expression of genes encoding effector T$_{reg}$ signatures such as *CD83*, *Pdcd1*, *Bcl6*, *Tnfrsf9*, *Irf4*, *Egr2*, *Cxcl10*, and *IL1r2*, TCR-inducing molecules such as *Nrn1*, *Slc7a10*, *Tnfsf11*, *CD38*, *Nt5e*, and *Plagl1*, and homing receptors *Ccr4*, *Ccr6*, and *Ccr9* was downregulated in Sharpin-deficient Treg (Fig. 1d). Furthermore, phenotypic analyses revealed that the percentage of effector Treg (CD44^{hi}CD62L^{lo}, CCR6+, or Ki67+) in the spleen and pLNs fell (Fig. 1e–g). In vivo co-transfer experiments using both naïve T cells and Treg revealed that Sharpin-deficient Treg failed to prevent recipient mice from developing T cell-dependent colitis and colitis-induced weight loss (Supplementary Fig. 1J). These data indicate that a sufficient amount of LUBAC is necessary to elicit signals that induce differentiation into an effector T$_{reg}$ phenotype and signals that regulate T$_{reg}$-mediated immune responses.

*Sharpin^{fl/fl}Foxp3^{Cre}* mice succumbed to chronic skin inflammation at around 4 weeks of age, and survived for at least 5 months (Fig. 1h). The skin lesions displayed autoinflammatory aspects similar to those observed in *cpdm* mice, which is a model of autoinflammatory dermatitis that shows hyperkeratosis, parakeratosis, keratinocyte apoptosis, lamellar fibrosis, and

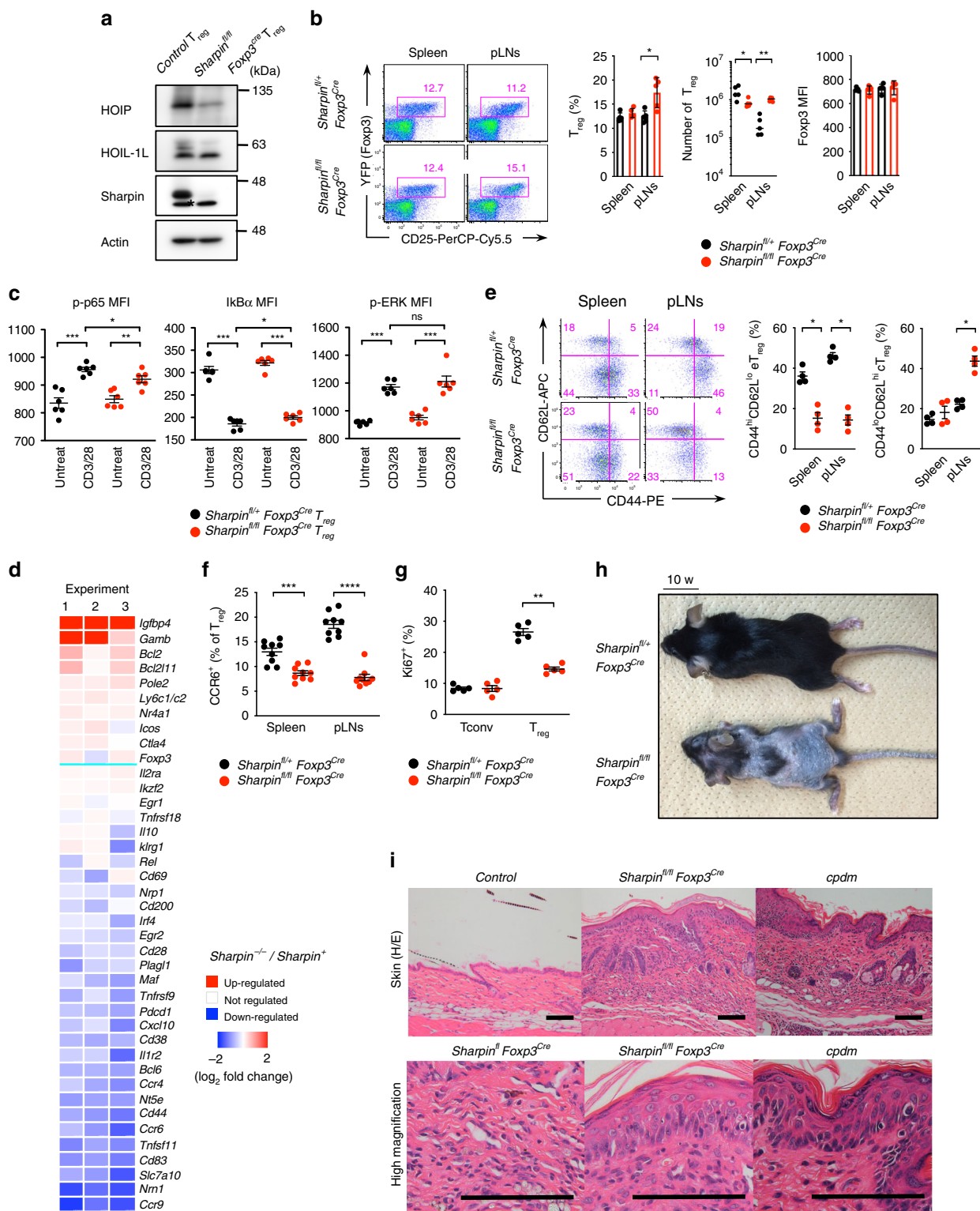

dermal infiltration by granulocytes (Fig. 1i). Histological analyses also revealed limited inflammation in the lungs, but the other tissues were normal (Supplementary Fig. 1K). They had lymphadenopathy, with marked influx of CD3+ T cells, B220+ B cells, and CD11b+ myeloid cells (Supplementary Fig. 1L, M). Thus, these results demonstrate that intrinsic loss of Sharpin impairs the immunosuppressive function of Treg by inhibiting differentiation into effector Treg, resulting primarily in chronic *cpdm*-like skin inflammation.

**T cells drive skin autoinflammation in *cpdm* mice.** Next, we generated *Sharpin*fl/fl*Lck*Cre mice that do not express Sharpin in most T cell lineages, including Treg and proinflammatory and cytotoxic T cells. A majority of *Sharpin*fl/fl*Lck*Cre mice exhibited no overt symptoms and survived normally, although around 12% suffered from dermatitis with generalized lymphadenopathy (Fig. 2a, b, Supplementary Fig. 2A). Other tissues appeared normal. Peripheral T cell lineages in healthy *Sharpin*fl/fl*Lck*Cre mice showed no significant changes, but a minor fraction acquired

**Fig. 1** Sharpin deficiency in Treg causes *cpdm*-like skin inflammation. **a** Western blot analysis of LUBAC components from sorted CD4$^+$CD25$^+$Foxp3$^+$ Treg. Actin is indicated as a loading control. Asterisks indicate non-specific bands. **b** Percentage of Foxp3$^+$CD25$^+$ Treg within the CD4$^+$ T cell population, absolute cell number of Treg, and the mean fluorescent intensity (MFI) of Foxp3 in Treg in the spleen and peripheral LNs (pLNs) from 10-week-old mice. $n = 5$ biologically independent animals. **c** MFI of signal mediators in Treg upon CD3/28 antibody-based activation. Intracellular staining was performed. Data for p-p65 or IκBα/p-ERK were acquired 5 and 10 min after activation, respectively. $n = 6$ biologically independent animals. **d** Expression of selected genes associated with T$_{reg}$ function and TCR-inducible T$_{reg}$ markers is presented in heat maps. Data are derived from three independent experiments. **e** Representative plots and percentage of effector (eT$_{reg}$; CD44$^{hi}$CD62L$^{lo}$) and central (cT$_{reg}$; CD44$^{lo}$CD62L$^{hi}$) T$_{reg}$ subsets in the spleen and pLNs. $n = 4$ biologically independent animals. **f** Percentage of CCR6$^+$ effector T$_{reg}$ subsets. $n = 9$ biologically independent animals. **g** Percentage of Ki67$^+$ T$_{reg}$ subsets. $n = 5$ biologically independent animals. **h** Appearance of skin inflammation at 10 weeks old. **i** Representative photos of H/E-stained skin of *Sharpin$^{fl/fl}$Foxp3$^{Cre}$*, control, and *cpdm* mice. Scale bar: 200 μm. Small circles in the graphs indicate data from an individual mouse. Small horizontal lines indicate the mean (±s.e.m.). ns, $p > 0.05$; *$p < 0.05$; **$p < 0.01$; ***$p < 0.001$; ****$p < 0.0001$. Two-tailed Mann–Whitney U-test was used for **b**, **c**, and **e–g**. Data are pooled from at least three independent experiments (**b**, **c**, **e–g**). Also, see Supplementary Fig. 1. Source data are provided in a Source Data file

activated (CD44$^{hi}$CD62L$^{hi/lo}$ or CD69$^{hi}$) phenotypes (Fig. 2c, d, Supplementary Fig. 2B–D). To probe defects in T cell immune responses in *Sharpin$^{fl/fl}$Lck$^{Cre}$* mice, we stimulated purified T cells in vitro. During the early phase of T cell activation, subtle inhibition of IκBα degradation was detected in Sharpin-deficient T cells, although expression of Nur77, an intermediate-early gene involved in TCR signaling, was comparable. In addition, at the late phase of activation we observed marked inhibition of CD25 and CD69 expression on the cell surface (Fig. 2g). Furthermore, an in vivo T cell cytotoxic assay revealed that *Sharpin*-deficient T cells induced T cell-mediated colitis and resultant weight loss (Fig. 2h, i); however, they had a lower pathogenic potency and exhibited a lower percentage of activated CD25$^+$ T cells in the mesenteric lymph nodes (Fig. 2h, j). We also found that depletion of B cells or Gr1$^+$ myeloid cells did not affect on the skin disease of *Sharpin$^{fl/fl}$Foxp3$^{Cre}$* mice (Supplementary Fig. 2E). Considering that development of autoimmune diseases depends largely on the immunological balance between self-reactive T cells and immunosuppressive Treg, these data suggest that the *cpdm*-like skin inflammation observed in *Sharpin$^{fl/fl}$Foxp3$^{Cre}$* mice is triggered, in some way, by T cell-mediated autoimmune pathogenesis.

Furthermore, to examine involvement of T cells in autoinflammatory skin pathogenesis in *cpdm* mice, we adapted a conditional expression system to express Sharpin along with EGFP (under control of the endogenous *ROSA26* promoter) in a Cre recombinase-dependent manner. We then generated genetically engineered *cpdm* mice in which Sharpin was specifically re-expressed in Treg or the Lck$^+$ T cell lineage (*cpdm R26-Sharpin; Foxp3$^{Cre}$* or *cpdm R26-Sharpin;Lck$^{Cre}$*, respectively) (Supplementary Fig. 3A). Recovered expression of Sharpin in these cells was detected by immunoblot or flow cytometry analysis (Supplementary Fig. 3B–D). Consistent with our findings in *Sharpin$^{fl/fl}$Foxp3$^{Cre}$* and *Sharpin$^{fl/fl}$Lck$^{Cre}$* mice, introduction of Sharpin into Treg delayed onset of dermatitis until after 3 months of age and did not induce growth disturbance (Fig. 3a, b). Histological analyses revealed suppressed epidermal hyperplasia and TUNEL-positive apoptotic cell death in the skin of *cpdm R26-Sharpin;Foxp3$^{Cre}$* and *cpdm R26-Sharpin;Lck$^{Cre}$* mice (Fig. 3c, d). The ameliorated skin inflammation led to marked suppression of autoinflammation-related cytokines such as TNFα, IL-1β, and IL-6, which were elevated in *cpdm* skin; similar observations were made with respect to development of immunopathological hallmarks of *cpdm* disease, such as splenomegaly and infiltration of multiple organs by myeloid cells (Fig. 3e–g and Supplementary Fig. 3E). In addition, the previously reported decrease in the T$_{reg}$ population in *cpdm* mice improved in these mice (Fig. 3h). These results indicate that normalization of T cell-mediated immunological functions improves skin autoinflammation in *cpdm* mice, and demonstrate that T cell-related mechanisms appear to play critical roles in multiple pathogeneses underlying skin autoinflammation of *cpdm* mice.

**Autoinflammation extrinsically downregulates Foxp3 in T$_{reg}$.** In contrast to *Sharpin$^{fl/fl}$Foxp3$^{Cre}$* mice, *cpdm* mice harbored a significantly lower population of peripheral Treg (Supplementary Fig. 4A)[20,21]. Since inflammation reduced expression of Foxp3 in Treg, we generated TNFα$^{−/−}$ *cpdm* mice to suppress progression of dermatitis. Suppression of TNFα-mediated dermatitis led to marked rescue of Foxp3 expression in T$_{reg}$, indicating that reducing the Foxp3$^+$ T cell population in *cpdm* mice was a secondary effect with respect to chronic dermatitis (Supplementary Fig. 4B). Furthermore, keratinocyte-specific Sharpin-deficient mice (*Sharpin$^{fl/fl}$K5$^{Cre}$*) developed *cpdm*-like dermatitis, suggesting that Sharpin expression by keratinocytes is requisite for tissue integrity, at least through inhibition of keratinocyte cell death (Supplementary Fig. 4C, D). Onset of the skin disease was independent on the presense of lymphocytes, but required for TNFα expression (Supplementary Fig 4I, J). *Sharpin$^{fl/fl}$K5$^{Cre}$* mice also displayed liver inflammation albeit to a lesser extent; however, other tissues were normal. Although Treg from *Sharpin$^{fl/fl}$K5$^{Cre}$* mice were genetically intact, they displayed reduced expression of Foxp3 and shifted to an activated and short-lived phenotype as dermatitis progressed (Supplementary Fig. 4E–G). In addition, experiments with thymic epithelial cell (TEC)-specific Sharpin-deficient mice (*Sharpin$^{fl/fl}$β5t$^{Cre}$*) revealed that expression of Sharpin in Keratin5$^+$ TECs was irrelevant with respect to T$_{reg}$ instability and development of dermatitis (Supplementary Fig. 4H). Collectively, these data indicate that chronic autoinflammation induced by keratinocyte cell death causes extrinsic suppression of Foxp3 in Treg that possibly elicited the previously reported autoimmune aspects of *cpdm* mice by augmenting the functions of effector T cells.

**Similar disease outcomes in *Sharpin$^{fl/fl}$Foxp3$^{Cre}$* and *K5$^{Cre}$*.** Involvement of T cells in development of *cpdm*-like skin disease in *Sharpin$^{fl/fl}$Foxp3$^{Cre}$* mice seems contradictory to the current concept of autoinflammatory disease. To compare immunopathological changes in *Sharpin$^{fl/fl}$Foxp3$^{Cre}$* with those in classical autoinflammatory disease, we used *Sharpin$^{fl/fl}$K5$^{Cre}$* mice in which augmented susceptibility to TNFα-induced keratinocyte cell death underlies dermatitis. The skin lesions in *Sharpin$^{fl/fl}$K5$^{Cre}$* mice resembled those in *cpdm*, including infiltration by F4/80$^+$ macrophages and MPO$^+$ granulocytes, but not lymphocytes (including T cells), which seems consistent with the context of autoinflammation (Fig. 4a). Next, we performed a detailed comparison of the immunological phenotypes of *Sharpin$^{fl/fl}$K5$^{Cre}$* and *Sharpin$^{fl/fl}$Foxp3$^{Cre}$* mice. Despite the T cell-mediated skin inflammation in *Sharpin$^{fl/fl}$Foxp3$^{Cre}$* mice, the skin lesions were infiltrated predominantly by myeloid cells (Fig. 4a, b). Both strains suffered from eosinophilia and neutrophilia, as defined by an increase in CD11b$^+$SiglecF$^+$ and CD11b$^+$Gr1$^{hi}$ cells, respectively, in the spleen (Fig. 4c). Moreover, expression of

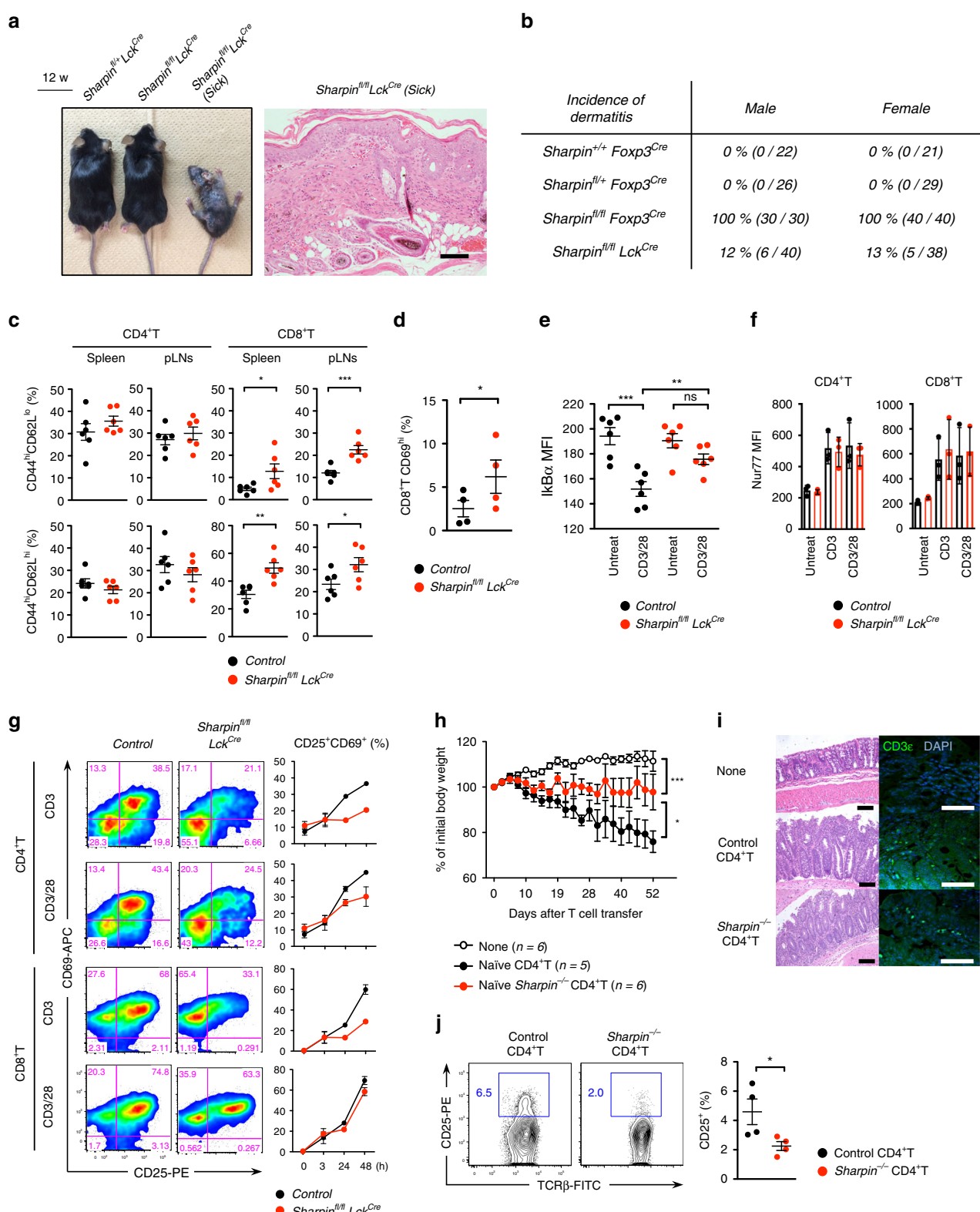

proinflammatory (IL-6, TNFα, and IL-1β) and allergy-related (TSLP) cytokines was very similar in $Sharpin^{fl/fl}Foxp3^{Cre}$ and $Sharpin^{fl/fl}K5^{Cre}$ mice (Fig. 4d). However, there were some differences between $Sharpin^{fl/fl}Foxp3^{Cre}$ and $Sharpin^{fl/fl}K5^{Cre}$ mice based on etiology. Peripheral granulocytes in $Sharpin^{fl/fl}K5^{Cre}$, but not $Sharpin^{fl/fl}Foxp3^{Cre}$, mice showed increased IL-6-mediated STAT3 phosphorylation, which is indicative of cytokine-dominant autoinflammatory traits (Fig. 4e). By contrast, $Sharpin^{fl/fl}Foxp3^{Cre}$, but not $Sharpin^{fl/fl}K5^{Cre}$, mice exhibited increased numbers of self-reactive CD69[+] T cells in peripheral tissues and the presence of skin-reactive autoantibodies in serum (Fig. 4f, g). In addition, only $Sharpin^{fl/fl}Foxp3^{Cre}$ mice showed distinct differences in progression of skin disease severity between males and females (gender-specific differences in severity are a common

**Fig. 2** Sharpin predisposes mice to spontaneous T cell-dependent dermatitis. **a** Representative appearance of 12-week-old control, healthy, and sick *Sharpin^fl/fl Lck^Cre* (left) mice and photos of H/E-stained skin (right). Scale bar: 200 μm. **b** Incidence of spontaneous dermatitis. All mice were monitored over a period of 6 months. **c** Percentage of effector CD44^hiCD62L^lo or memory CD44^hiCD62L^hi subsets within the CD4+ or CD8+ T cell populations in the spleen and pLNs from 12-week-old healthy *Sharpin^fl/fl Lck^Cre* mice. $n = 6$ biological independent animals. **d** Percentage of recently activated CD69+ cells within the CD8 T cell population in pLNs. $n = 4$ biological independent animals. **e** MFI of IκBα in T cells at 10 min after CD3/28 antibody-based TCR stimulation. Intracellular staining was performed. $n = 6$ biological independent animals. **f** MFI of Nur77 in T cells at 3 h after CD3 or CD3/CD28 antibody-based TCR stimulation. $n = 3$ biological independent animals. **g** CD69 and CD25 expression by CD4+ or CD8+ T cells at the indicated times after TCR stimulation. **h** Experimental induction of colitis by naïve CD4+ T cells. *Rag2^−/−* mice were intravenously injected with HBSS alone or with $2.5 \times 10^5$ sorted naïve CD4+ T cells (CD45RB^hiCD25−CD4+CD3ε+), which were obtained from pooled cells isolated from the spleen and pLNs of 4-week-old mice. Bodyweight of injected *Rag2^−/−* mice was monitored twice or thrice weekly. Data indicate average values. **i** Representative photos of H/E and immunofluorescent staining against CD3+ T cells within the inflamed colon of mice in **h**. **j** Percentage of CD25+CD4 T cells in the mesenteric LN (mLN) of mice in **h**. $n = 4$ biological independent animals. Small horizontal lines indicate the mean (±s.e.m.). ns, $p > 0.05$; *$p < 0.05$; **$p < 0.01$; ***$p < 0.001$. Kruskal–Wallis test with Bonferroni correction (α value = 0.05) was used for **h**, while two-tailed Mann–Whitney U-test was used for **c–f** and **j**. Data are pooled from at least three independent experiments (**c–g**, **j**). Also, see Supplementary Fig. 2. Source data are provided in a Source Data file

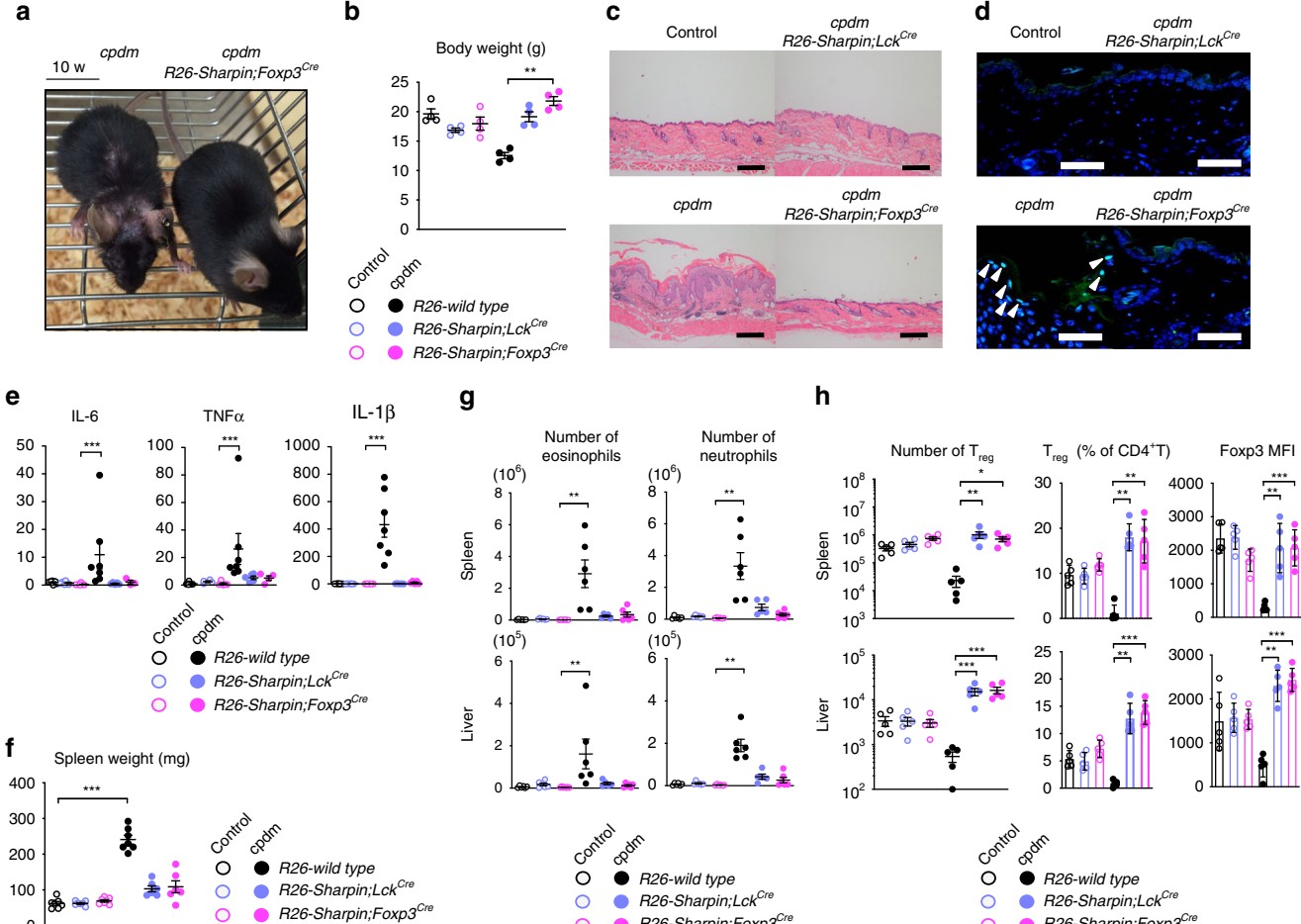

**Fig. 3** Genetic compensation of Sharpin in T lineages rescues *cpdm* mice from autoinflammatory skin disease. **a** Representative appearance of 10-week-old *cpdm* and *cpdm R26-Sharpin;Foxp3^Cre* mice. **b** Body weight of indicated 10–12-week-old mice. $n = 4$ biologically independent animals. **c, d** Representative skin sections from the indicated 12-week-old mice. Scale bars: 200 μm for H/E (**c**) and 50 μm for TUNEL staining (**d**). TUNEL-positive epidermal cells are indicated as arrowheads. **e** Quantitative analyses of mRNAs encoding inflammation-related cytokines in the skin of 10–12-week-old mice. **f** Spleen weight from the indicated 10–12-week-old mice. $n = 7$ biologically independent animals. **g** Absolute number of eosinophils (CD11b+Gr1+SiglecF+) and neutrophils (CD11b+Gr1^hi) in the spleen (upper) and liver (lower). $n = 6$ biologically independent animals. **h** Absolute number and percentage of Treg, and MFI of Foxp3 in Treg, in the spleen and liver of 10–12-week-old mice. $n = 5$ biologically independent animals. Data are representative of two independent experiments (**c, d**) or pooled from three independent experiments (**b, e–h**). Small circles indicate individual mice. Small horizontal lines indicate the mean (± s.e.m.). *$p < 0.05$, **$p < 0.01$, ***$p < 0.001$. Kruskal-Wallis test with Bonferroni correction (α value = 0.05) was used for **b, e–h**. Also, see Supplementary Fig. 3. Source data are provided in a Source Data file

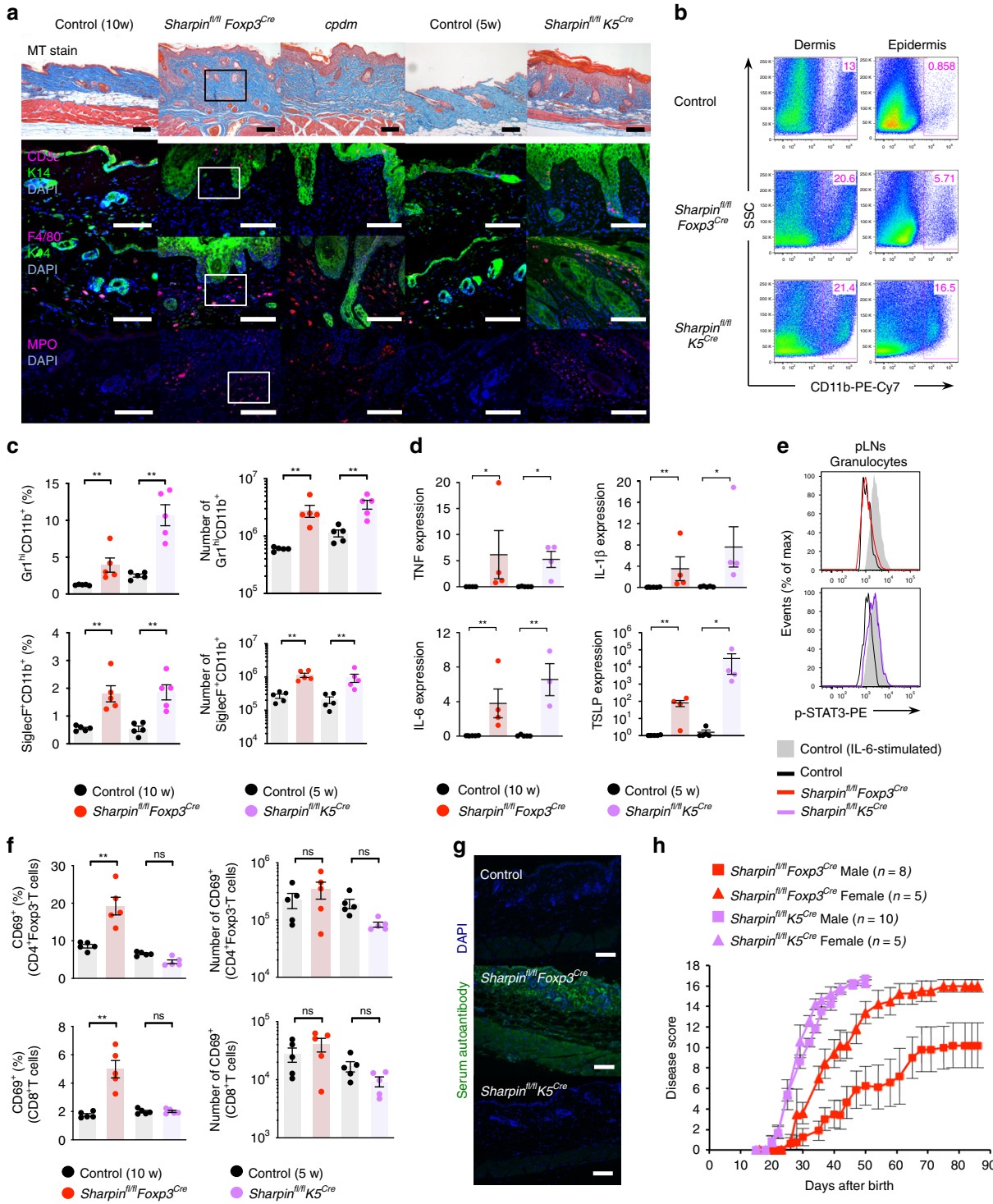

**Figure panels a–h**

characteristic of autoimmune diseases) (Fig. 4h). Thus, the data indicate that T cell-induced autoinflammation is a main pathogenesis underlying T cell-elicited immunological changes in *Sharpin^{fl/fl}Foxp3^{Cre}* mice. However, the comparable skin pathologies suggest that pathological changes in *Sharpin^{fl/fl}Foxp3^{Cre}* mice are provoked in a manner similar to those in autoinflammatory *Sharpin^{fl/fl}K5^{Cre}* mice.

**Skin-oriented T cells induce programmed keratinocyte death.** Next, we examined autoinflammatory manifestations in

*Sharpin^{fl/fl}Foxp3^{Cre}* mice in more detail. Keratinocyte death is a critical event that triggers autoinflammation in *cpdm* mice since LUBAC protects against TNFα-induced caspase-dependent apoptosis and necroptosis; the latter is a type of programmed necrotic cell death mediated by RIPK1, RIPK3, and MLKL[23]. Immunohistochemical staining of cleaved caspase-3 (apoptosis) or phosphorylated MLKL (necroptosis) confirmed that both types of cell death occur in keratinocytes in the inflamed skin of *Sharpin^{fl/fl}K5^{Cre}* mice (Fig. 5a, b). To our great interest, both types of cell death were also observed in the skin

**Fig. 4** Distinct etiologies, autoimmunity in *Sharpin^{fl/fl}Foxp3^{Cre}* or autoinflammation in *Sharpin^{fl/fl}K5^{Cre}*, result in similar myeloid cell-dominant inflammatory outcomes. **a** Pathology underlying skin inflammation. Skin sections were subjected to Masson's trichrome and immunofluorescence staining. The types of infiltrating immune cell were identified by staining with antibodies specific for CD3ε (T cells), F4/80 (macrophages), and MPO (neutrophils). Keratin14 (K14) was used as a structural marker of the epidermis. The outlined areas are magnified in Fig. 7f. Scale bars: 100 μm. **b** Percentage of skin-infiltrating CD11b^+ cells. **c** Representative plots, percentages, and absolute numbers of eosinophils (CD11b^+SiglecF^+) and neutrophils (CD11b^+Gr1^{hi}) in the spleen in 6-week-old *Sharpin^{fl/fl}K5^{Cre}* and 10-week-old *Sharpin^{fl/fl}Foxp3^{Cre}* and respective control mice. $n = 5$ biologically independent animals. **d** Quantitative analysis of skin mRNA encoding inflammation-related cytokines. $n = 4$ biologically independent animals. **e** Histogram showing the presence of phosphorylated STAT3 in Gr1^+ granulocytes isolated from secondary lymph nodes. As a positive control, cells were incubated for 15 min at 37 °C with IL-6 (30 ng/ml) prior to intracellular staining. **f** Percentage of activated CD69^+ subsets in CD4^+ or CD8^+ T cells in secondary lymphoid tissues. $n = 5$ biologically independent animals. **g** Detection of skin-associated autoantibodies in serum. Scale bars: 100 μm. **h** Severity scores for spontaneous skin inflammation. Each score represents an average value from the indicated number of mice. Small horizontal lines indicate the mean (±s.e.m.). ns, $p > 0.05$; *$p < 0.05$; **$p < 0.01$. Two-tailed Mann–Whitney U-test was used for **c**–**f**. Data are representative of at least three independent experiments (**a**, **b**, **e**, **g**), or pooled from two independent experiments (**c**, **d**, **f**). Also, see Supplementary Fig. 4. Source data are provided in a Source Data file

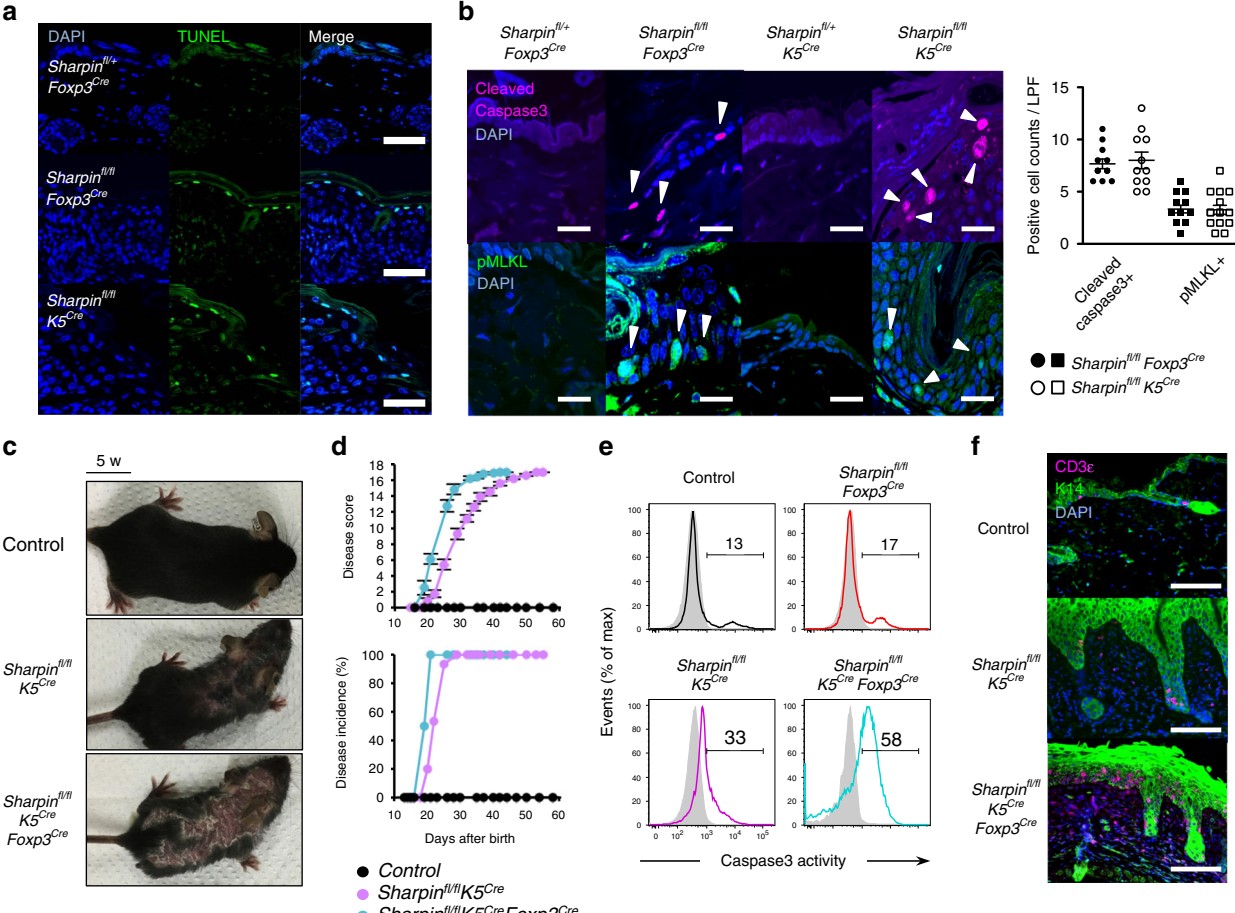

**Fig. 5** Skin-oriented T cells induce apoptotic and necroptotic keratinocyte death in an inflammatory setting. **a** TUNEL staining of skin sections from 5-week-old *Sharpin^{fl/fl}K5^{Cre}*, 10-week-old *Sharpin^{fl/fl}Foxp3^{Cre}*, and control mice. Scale bars: 50 μm. **b** Immunofluorescence staining of skin sections. Cleaved caspase-3 and phosphorylated MLKL were used as markers of apoptosis and necroptosis, respectively. Scale bars: 20 μm. Cell death in the epidermis was quantified by counting the total number of bright cells per low-power field (LPF). **c** Appearance of skin dermatitis. **d** Progression (upper) and incidence (lower) of skin disease among the indicated strains (numbers of mice: control, $n = 12$; *Sharpin^{fl/fl}K5^{Cre}*, $n = 15$; *Sharpin^{fl/fl}K5^{Cre}Foxp3^{Cre}*, $n = 8$). Disease scores were recorded up to 60 days after birth. No gender-based differences in the incidence of skin disease were observed. **e** Representative histogram showing death-prone epidermal cells from the indicated strains. Isolated epidermal cells were incubated with PhiPhiLux-G_1D_2, a fluorescent substrate of active caspase-3. **f** Immunofluorescence staining of skin sections to detect skin-infiltrating T cells. Scale bars: 100 μm. The small horizontal lines in **b**, **d** indicate the mean (±s.e.m.). Data are representative of at least three independent experiments. Source data are provided in a Source Data file

of *Sharpin^{fl/fl}Foxp3^{Cre}* mice; indeed, the phenomena were almost as common in these mice as in *Sharpin^{fl/fl}K5^{Cre}* mice (Fig. 5a, b). Additional elimination of Sharpin from Treg in *Sharpin^{fl/fl}K5^{Cre}* mice (*Sharpin^{fl/fl}K5^{Cre}Foxp3^{Cre}*) exacerbated skin inflammation, with a marked increase in caspase-3 activity in CD45^{neg} keratinocytes, which led to earlier onset of disease

(Fig. 5c–e). Moreover, a massive T cell infiltrate was observed in the skin of *Sharpin^{fl/fl}K5^{Cre}Foxp3^{Cre}* mice (Fig. 5f). Collectively, these results strongly suggest that skin-oriented activated T cells promote keratinocyte death and induce skin autoinflammation, and that induction of cell death underlies T cell-mediated inflammatory processes.

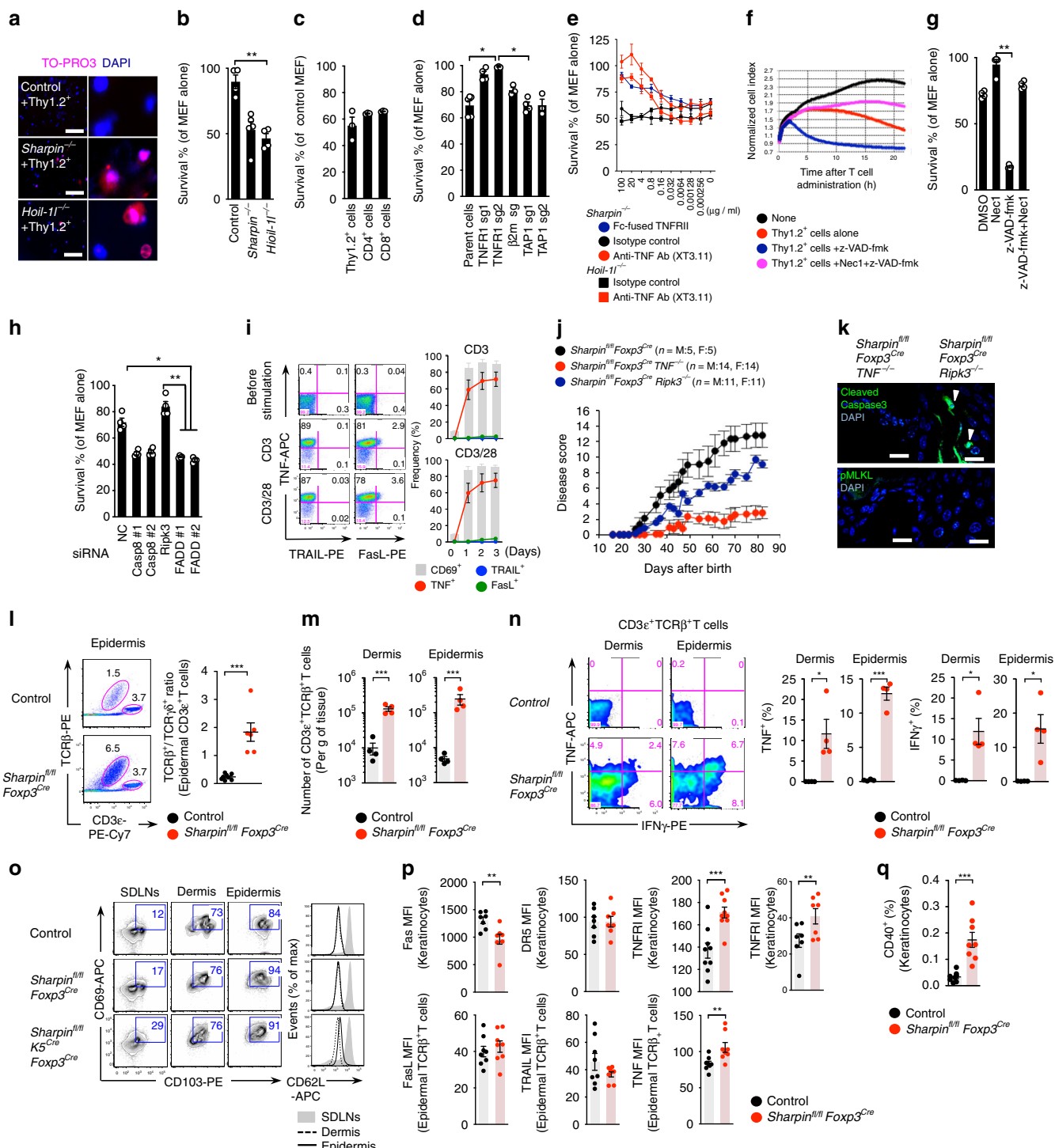

**Fig. 6** A pathogenic role of the TNFα-TNFRI axis during T cell-mediated autoinflammation. **a** Staining of MEFs co-cultured with T cells. Scale bars: 100 μm. Magnified images are shown on the right. **b–d** Percentage of surviving cells in **a** (**b**), $Sharpin^{-/-}$ MEFs co-cultured with the indicated T cells (**c**), and the indicated CRISPR/Cas9-targeted MEFs (**d**) $n = 4$ biologically independent experiments. **e** Treatment with a TNFα-neutralizing antibody or solubilized TNFRII. **f** Treatment with z-VAD-fmk and/or Nec1. The iCelligence system was used to measure cell viability. **g, h** Survival after 10 h of the treatment described in **f** (**g**) or after siRNA knockdown (**h**). $n = 4$ independent experiments. **i** Expression of the indicated receptors on the T cell surface after TCR stimulation. **j** Disease progression among the indicated strains. M, Male; F, Female. **k** Immunofluorescence staining of skin sections. Scale bars: 20 μm. **l** Ratio of $CD3\varepsilon^{int}TCR\beta^{+}$:$CD3\varepsilon^{hi}TCR\gamma\delta^{+}$ cells. $n = 6$ biologically independent animals. **m** Number of $TCR\beta^{+}$ T cells in the skin. $n = 4$ biologically independent animals. **n** Percentage of TNFα- and/or IFNγ-expressing $TCR\beta^{+}$ T cells in the skin. $n = 4$ biologically independent animals. **o** Dot plots representing $T_{RM}$ cells. **p, q** MFI or percentage of cells expressing the indicated surface proteins. $n = 8$ biologically independent animals. Small horizontal lines indicate the mean (±s.e.m.). *$p < 0.05$, **$p < 0.01$, ***$p < 0.001$. Kruskal–Wallis test with Bonferroni correction (α value = 0.05) was used for **b–d**, **g**, and **h**, while two-tailed Mann–Whitney U-test was used for **l–n**, **p**, and **q**. Data are representative of at least two independent experiments (**a**, **f**, **i**, **k**, **o**), or pooled from three to six (**b–e**, **g**, **h**, **l–n**, **p**, **q**) independent experiments. Source data are provided in a Source Data file

**TNFα-TNFRI axis implicates T cell-triggered autoinflammation.** To examine the molecular mechanisms underscoring T cell-mediated autoinflammation, we set up co-culture experiments with primary T cells as effectors and fibroblasts as targets. In these experiments, 40–50% of LUBAC-compromised MEFs died within 10 h (Fig. 6a, b). Cell death was not restricted by MHC because both isolated CD4$^+$ and CD8$^+$ T cells induced death at comparable levels. Instead, deletion of *Tnfrsf1a* genes encoding TNFRI, but not *β2m* or *Tap1*, protected target cells from death (Fig. 6c, d). Furthermore, a TNFα-neutralizing antibody (XT3.11) and a soluble Fc-fused TNFRII protein (etanercept) inhibited T cell-dependent cell death in a dose-dependent manner (Fig. 6e). TNFα is a well-known inducer of apoptosis and/or necroptosis under certain conditions[24,25]. The pan-caspase inhibitor z-VAD-fmk (z-VAD) and the RIPK1 kinase inhibitor Necrostatin-1 (Nec1) inhibit apoptosis and necroptosis, respectively. Treatment with z-VAD alone sensitized MEFs to necroptosis, whereas concomitant treatment with both z-VAD and Nec1 rescued them from death almost completely, even in long-term co-culture (Fig. 6f, g). These data were confirmed by the observation that target cells treated with siRNA targeting the necroptosis regulator RIPK3, but not apoptosis regulators such as caspase-8 or FADD, showed improved survival (Fig. 6h). Moreover, considering that the unstimulated T cells used herein showed only weak expression of TNFα, and that expression increased preferentially in most T cells after TCR stimulation (Fig. 6i), we anticipated that TNFα would be an exacerbating factor for T cell-related inflammatory diseases. Therefore, to determine whether TNFα-induced programmed cell death plays a role in the autoinflammatory-like pathogenesis in *Sharpin$^{fl/fl}$Foxp3$^{Cre}$* mice, we generated *Sharpin$^{fl/fl}$Foxp3$^{Cre}$* mice lacking TNFα or RIPK3. We found that both mouse strains showed clear amelioration of skin disease progression with restricted occurrence of programmed cell death (Fig. 6j, k). Indeed, the number of TCRβ$^+$, but not TCRγδ$^+$, T cells in the skin of *Sharpin$^{fl/fl}$Foxp3$^{Cre}$* mice was more than 10-fold higher than that in control mice, and skin TCRβ$^+$ cells expressed much more TNFα and IFNγ, although they were a minor component of the skin infiltrate (Fig. 6l–n). It was interesting that most of the skin-infiltrating T cells appeared to have a CD103$^+$CD69$^+$CD62L$^{lo}$ resident memory T (T$_{RM}$) phenotype because T$_{RM}$ cells reside long-term and function protectively in the skin (Fig. 6o)[26,27]. Consistent with the results of in vitro T cell cytotoxicity studies, the TNFα-TNFRI axis, rather than other necroptosis-inducible death signaling pathways via DR5 or Fas receptor, appeared primarily to drive keratinocyte cell death in *Sharpin$^{fl/fl}$Foxp3$^{Cre}$* mice (Fig. 6p). Furthermore, we observed upregulated expression of TNFRII and CD40, a receptor for T cell activation co-stimulator CD40L, on keratinocytes (Fig. 6p, q)[28–31]. Thus, these in vivo and in vitro experiments confirm the pathogenicity of multiple types of T cell-mediated programmed cell death and reveal the underlying role of the TNFα-TNFR1 axis during skin autoinflammation.

**T cell-triggered cell death modulates autoimmune outcomes.** We clarified that T$_{reg}$-specific ablation of *Sharpin* preferentially induces T cell-mediated autoinflammation, whereas loss of *Hoip* from Treg, in which the LUBAC complex no longer exists, causes a fatal multi-organ inflammatory response that closely resembles the pathology in *Foxp3*-lacking *Scurfy* mice, a classical model of autoimmune disease[32]. Taking advantage of the unique characteristics of LUBAC (i.e., loss of each LUBAC subunit affects the amount of the complex to a different extent), we generated HOIP conditional knockout mice to establish three engineered mouse strains harboring Treg expressing different levels of LUBAC (*Sharpin$^{fl/fl}$Foxp3$^{Cre}$* > *Sharpin$^{fl/fl}$Hoip$^{fl/+}$Foxp3$^{Cre}$* > *Hoip$^{fl/fl}$Foxp3$^{Cre}$*) (Supplementary Fig. 5A, B). This enabled us to compare diseases provoked by impairment of Treg expressing different levels of a single component of the functional complex (Fig. 7a). Intriguingly, these mice showed quite distinct inflammatory outcomes. Comparative analyses of the three mouse strains revealed a gradual reduction in the proportion and absolute number of Treg in accordance with the reduced levels of LUBAC (Supplementary Fig. 5C). Although expression of T$_{reg}$ functional markers did not change, expression of Nrp1 and Helios by Treg fell markedly upon *Hoip* ablation (Supplementary Fig. 5D, E). In accordance with the reduction of LUBAC in Treg, the disease appeared to worsen and the number of inflamed organs increased (Fig. 7b, Supplementary Fig. 5F, G). In addition to the skin and lung, the pancreas in *Sharpin$^{fl/fl}$Hoip$^{fl/+}$Foxp3$^{Cre}$* mice, along with all tissues investigated in *Hoip$^{fl/fl}$Foxp3$^{Cre}$* mice, were affected. We also observed increased proportions of activated (CD44$^{hi}$CD62L$^{lo}$, GITR$^{hi}$, CD25$^{hi}$, or CD69$^+$) T cells and an increase in Th2-dominant inflammation, which is characterized by increased production of cytokines (IL-4, IL-5, IL-10, and IL-13 from CD4$^+$ T cells) and by secretion of IgG1 and IgE antibodies into the serum (Fig. 7c, d, Supplementary Fig. 5H, I). In contrast to *Sharpin$^{fl/fl}$Foxp3$^{Cre}$* skin lesions, massive T cell infiltrates were observed in *Hoip$^{fl/fl}$Foxp3$^{Cre}$* lesions (Fig. 7e, f). Importantly, deletion of one allele of the *Hoip* locus in *Sharpin$^{fl/fl}$Foxp3$^{Cre}$* (*Sharpin$^{fl/fl}$Hoip$^{fl/+}$Foxp3$^{Cre}$*) mice rendered skin inflammation more severe and, intriguingly, the T cell-poor lesions in *Sharpin$^{fl/fl}$Foxp3$^{Cre}$* mice were converted to a T cell-dominant type in *Sharpin$^{fl/fl}$Hoip$^{fl/+}$Foxp3$^{Cre}$* mice, which can be regarded as an autoimmune disease (Fig. 7e–g). Moreover, it is noteworthy that epithelial cells in *Sharpin$^{fl/fl}$Hoip$^{fl/+}$Foxp3$^{Cre}$* mice underwent apoptotic and necroptotic cell death, as observed in *Sharpin$^{fl/fl}$Foxp3$^{Cre}$* mice (Fig. 7h). Since the various types of inflammation observed among the three strains were clearly T cell-dependent, the pathologies of T cell-mediated autoimmune diseases are generally modulated by uncovered T cell-triggering innate inflammatory responses.

## Discussion

Elimination or suppression of T$_{reg}$-mediated peripheral tolerance is sufficient to induce T cell-triggered immunological disorders, which are categorized as autoimmune diseases[33]. Here, we performed comprehensive analysis of mice bearing genetic mutations of subunits of the trimetric LUBAC ubiquitin ligase in Treg and demonstrated that impairment of Treg triggers diverse pathological manifestations. Mice lacking HOIP, a catalytic subunit of LUBAC, in Treg exhibited classical autoimmune disease accompanied by massive lymphocyte infiltration. However, mice lacking an accessory Sharpin subunit specifically in Treg developed a quite distinct form of inflammation, which is an apparently T cell-poor but innate immune cell-dominant inflammatory response, although they still exhibited some autoimmune traits such as activated peripherally circulating T cells and autoantibody production. These results demonstrating the pathological diversity of T cell-mediated sterile inflammatory diseases appear to contradict the current immunopathological definition, i.e., that self-reactive T cells can be the main driver of autoimmunity but not autoinflammation. Our further analyses revealed that loss of an additional *Hoip* locus in mice harboring Treg lacking *Sharpin* (*Sharpin$^{fl/fl}$Hoip$^{fl/+}$Foxp3$^{Cre}$*) resulted in more severe skin inflammation in which both massive T cell infiltration and

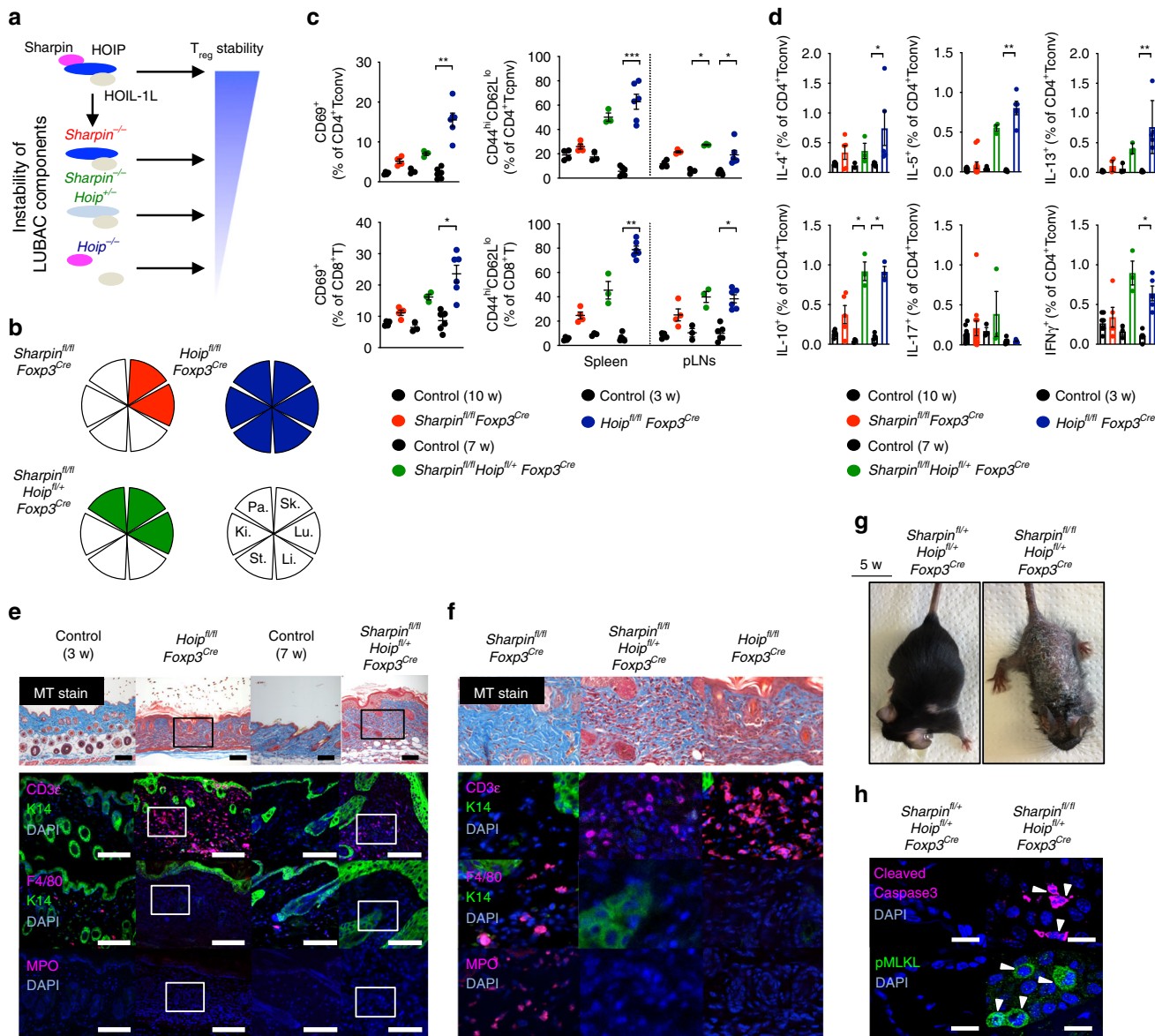

**Fig. 7** T cell-dependent autoinflammation is a pathogenesis common to several autoimmune diseases. **a** Schematic diagram showing genetic targeting of LUBAC, leading to impairment of both TCR-mediated NF-κB signaling in Treg and T$_{reg}$ stability. **b** Tissues affected during chronic inflammation. Sk, Skin; Lu, Lung; Li, Liver; St, Stomach; Ki, Kidney; Pa, Pancreas. **c** Percentage of CD44$^{hi}$CD62L$^{lo}$ effector subsets or CD69$^+$ recently activated subsets within the CD4+Foxp3$^-$ or CD8$^+$ T cell populations, respectively. *Sharpin$^{fl/fl}$Foxp3$^{Cr}$*, $n = 4$; *Sharpin$^{fl/fl}$Hoip$^{fl/+}$Foxp3$^{Cre}$*, $n = 3$; *Hoip$^{fl/fl}$Foxp3$^{Cre}$*, $n = 6$ biologically independent animals. **d** Percentage of cytokine-producing CD4$^+$ T cells. CD4$^+$ T cells isolated from secondary lymphoid tissues were stimulated with PMA and ionomycin for 4 h before intracellular staining for each cytokine. *Sharpin$^{fl/fl}$Hoip$^{fl/+}$Foxp3$^{Cre}$*, $n = 3$; the other strains, $n = 5$ biologically independent animals. **e** Pathology underlying skin inflammation. Skin sections were subjected to Masson's trichrome and immunofluorescence staining. Keratin14 (K14) was used as a structural marker of the epidermis. Scale bars: 100 μm. The outlined areas are magnified in **f**. **f** Magnified photos from **e** and Fig. 4g. **g** Representative appearance of skin inflammation. Inflammation in 5-week-old *Sharpin$^{fl/fl}$Hoip$^{fl/+}$Foxp3$^{Cre}$* was more severe than that in *Sharpin$^{fl/fl}$Foxp3$^{Cre}$* mice. **h** Immunofluorescence staining of programmed cell death in skin from the indicated mice. Scale bars: 20 μm. Small circles indicate individual mice. Small horizontal lines indicate the mean (±s.e.m.). *$p < 0.05$, **$p < 0.01$, ***$p < 0.001$. Kruskal-Wallis test with Bonferroni correction ($\alpha$ value $= 0.05$) was used for **c** and **d**. Data are representative of at least three independent experiments (**b**, **e**, **f**, **h**) or pooled from three independent experiments (**c**, **d**). Also, see Supplementary Fig. 5. Source data are provided in a Source Data file

predominant keratinocyte cell death developed. Considering the apparent innate inflammatory changes in *Sharpin$^{fl/fl}$Foxp3$^{Cre}$* mice (the strain showing the most mild impairment of Treg), the results indicated that self-reactive T cells can readily provoke innate inflammatory insults through T cell-mediated cytotoxicity rather than typical autoimmune responses.

TNFα, a well-known master proinflammatory cytokine, is produced mainly by T cells, NK cells, monocytes, and activated macrophages, and functions as a homotrimer complex of membrane-bound or secreted forms to regulate immune-related physiological processes or diseases. Here, we showed clear and significant upregulation of TNFα on the CD4$^+$ and CD8$^+$ T cell

surface in response to TCR-mediated stimulation, along with its direct association with death of epidermal cells in mice. Our co-culture system revealed TNFα-induced T cell-mediated pro-grammed cell death (Fig. 6d, e); therefore, upregulated TNFα should be a primary inducer of cell death mediated by self-reactive T cells. However, in our assays, MEFs and primary keratinocytes from wild-type mice were not killed effectively in a TNFα-dependent manner, even though they were co-cultured with activated T cells. This implies that secondary signaling mediated by inflammatory cytokines may be essential for induction of TNFα-TNFRI-mediated cell death in a physiological setting. For example, simultaneous activation of TNFRII expressed by target cells might play a role since we detected upregulated TNFRII expression on keratinocytes from *Sharpin^fl/fl^Foxp3^Cre^* mice. Secreted TNFα activates TNFRI preferentially, whereas membrane-bound TNFα, expression of which is enforced in activated T cells, strongly interacts with both TNFRI and TNFRII to induce distinct signals. Unlike TNFRI, TNFRII cannot induce cell death as it has no cytoplasmic death domain[34]. However, upon binding to TNFα, TNFRII recruits TRAF2 and TRAF2-associated protective factors, such as cIAP1 and cIAP2, which may result in reduced recruitment of these signaling molecules to TNFRI, leading to enhanced induction of TNFRI-mediated cell death[29,35,36]. Also, other surface molecules, such as CD40L, expressed on activated effector T cells via an as-yet-undetermined mechanism may cooperate to facilitate TNFRI-mediated necroptotic cell death[28,30].

The TNFα-TNFRI axis induces two types of programmed cell death under certain conditions: caspase-dependent apoptosis and RIPK1-RIPK3-MLKL-dependent necroptosis. In this context, we detected both types in the inflamed epidermis of *Sharpin^fl/fl^Foxp3^Cre^* mice. Necroptosis induces necrosis-like morphological disruption and subsequently releases intracellular factors, such as DAMPs, which can exacerbate inflammation. Also, recent studies implicate necroptosis in the pathogenesis of inflammatory bowel disease, systemic lupus erythematosus, and multiple sclerosis[37,38]. TNFα-induced cell death in an in vitro co-culture system was suppressed effectively by inhibiting necrop-tosis (Fig. 6h), which led us to attempt deletion of *RIPK3* from *Sharpin^fl/fl^Foxp3^Cre^* mice (*Sharpin^fl/fl^Foxp3^Cre^RIPK3^-/-^*) to assess the contribution of necroptosis to the pathogenesis of innate inflammation. Amelioration of dermatitis via loss of RIPK3 indicated a significant role for RIPK3-mediated necroptosis in driving T cell-induced innate inflammation. However, apoptosis may also contribute to some extent because loss of RIPK3 failed to fully cure dermatitis in *Sharpin^fl/fl^Foxp3^Cre^* mice (Fig. 6j, k). These data may support previous reports showing that deleting either *RIPK3* or *MLKL* does not rescue *cpdm* mice from autoinflammatory skin disease completely, whereas *RIPK3^-/-^Caspase-8^+/-^ cpdm* mice do not show overt symptoms of dermatitis[13,14]. As shown in *Sharpin^fl/fl^K5^Cre^Foxp3^Cre^* mice, overactivation of self-reactive T cells in autoinflammatory settings induces apoptosis in most keratinocytes (Fig. 5f). Thus, apoptosis induced by T cell-expressed TNFα in the skin might exhibit inflammatory potential, although apoptosis is generally regarded as a non-immunogenic form of cell death. It is unclear how T cells would induce different types of cell death in vivo, or which forms of cell death would contribute to different inflammatory conditions. Therefore, further detailed studies are needed if we are to gain better understanding of the mechanisms underlying T cell-induced inflammation; such studies may lead to effective therapeutic strategies for T cell-mediated (but apparently T cell-poor) systemic inflammatory diseases.

In summary, we provide strong evidence that autoimmune-prone T cells can initiate innate inflammation via TNFα-induced cell death; as far as we are aware, this is a new concept with respect to immune-mediated inflammatory mechanisms. Con-sistent with our conclusions, TNFα plays crucial roles not only in human autoinflammatory diseases but also in autoimmune dis-eases such as rheumatoid arthritis, psoriasis, and Crohn's disease. In addition, based on our observation of apparent pathological conversion of innate inflammatory skin in *Sharpin^fl/fl^K5^Cre^* mice into autoimmune skin in *Sharpin^fl/fl^K5^Cre^Foxp3^Cre^* mice, we suggest that TNFα-mediated cytotoxicity via slightly activated T cells may trigger innate-type inflammation, which is involved in surprising diverse pathological outcomes depending on the environment within the affected tissue. Thus, aberrant T cell immune tolerance could induce autoimmunity and/or innate inflammatory mechanisms that lead to complex T cell-mediated immunopathophysiology. Our findings may explain the para-doxical etiological outcomes and the propounded continuum model for development of autoimmune and autoinflammatory disorders[39–41].

## Methods

**Mice.** All animal use and care were performed according to protocols approved by the Animal Research Committee, Graduate School of Medicine, Kyoto University, and we complied with all the ethical regulations. All mice were housed at the Institute of Laboratory Animals, Kyoto University, under specific pathogen-free conditions, and all animal experiments were conducted in accordance with the guidelines for animal experiments at Kyoto University and RIKEN Kobe Branch. C57BL/6 (B6) mice were obtained from SLC, Japan. *Cpdm* mice were described previously[7]. *Rag2^-/-^* mice were obtained from the Central Institute of Experi-mental Animals, Japan. *TNFα^-/-^* (Stock number: 003008) and B6 (CD45.1) (Stock number: 002014) mice were obtained from the Jackson Laboratory. Lck-Cre (Model number: 4197) transgenic mice were obtained from Taconic. β5t-iCre knock-in mice and *Foxp3-YFP/iCre* knock-in mice were kindly provided by You-suke Takahama (University of Tokushima, Japan) and Alexander Y. Rudensky (Memorial Sloan Kettering Cancer Center, USA), respectively[42,43]. *Ripk3^-/-^* mice were kindly provided by Vishva Dixit (Genentech, USA)[44]. K5-Cre mice were obtained from CARD-Kumamoto University, Japan[45]. Sharpin conditional KO, HOIP conditional KO, and Sharpin conditional transgenic mice were generated in-house, and are available from RIKEN BRC, Japan.

**Generation of conditional Sharpin-KO mice.** A targeting vector for floxed *Sharpin* mutant mice was constructed by inserting loxP sequences bracketing exons 3–9 of the *Sharpin* gene, which encodes the ubiquitin-like (UBL) domain and a NZF domain of the Sharpin protein and is deleted after expression of Cre recombinase. A neomycin resistance gene flanked by FRT sites was inserted into intron 2. Bruce 4 C57BL/6 embryonic stem (ES) cells were transfected with the targeting vector and screened as neomycin-resistant colonies. Appropriate homo-logous recombination was confirmed by Southern blot analysis. The targeted ES cells were injected into recipient blastocysts to generate germline-transmitting chimeras. Mice carrying the mutated *Sharpin* locus were crossed with *Flp* Deleter mice to delete the neomycin cassette and then backcrossed onto the C57BL/6N strain for at least five generations before analysis. The following PCR primers were used to genotype the conditional Sharpin-KO mice: Forward (Fw-Intron2), GTGACAAGGTCTCAATGTGAAT; Reverse (Rv-Intron2), CTGTAATCCCAGT GTTCATATG; and Reverse (Rv-CYC1), TCCATGGCCTTCTCAGGCC. Size of floxed allele: 500 bp; size of wild-type allele: 370 bp.

**Generation of conditional HOIP-KO mice.** A targeting vector for floxed *Hoip* mutant mice was constructed by inserting loxP sequences bracketing exons 7–11 of the *Hoip* gene, which encodes the zinc finger domains and ubiquitin-associated (UBA) domain of the HOIP protein and is deleted after expression of Cre recombinase. A neomycin resistance gene flanked by FRT sites was then inserted into intron 6. JM8 C57BL/6 N ES cells were then transfected with the targeting vector. The following steps were the same as those used to generate conditional Sharpin-KO mice. The PCR primers used to genotype the conditional HOIP-KO mice were as follows: Forward (fl–f4), ACACTAAGCAAGTAGCAGCCAC; and Reverse (fl–r5), GCTCTGGCCTTTCTGATTGCTG. Size of floxed allele: 370 bp; size of wild-type allele: 220 bp.

**Generation of conditional Sharpin transgenic mice.** A targeting vector for conditional Sharpin transgenic mice was constructed by insertion of the *Sharpin* sequence fused to an N-terminal FLAG-His tag, followed by the FRT-flanking IRES-EGFP cassette plus the polyA sequence, into the *Rosa26* locus. The neomycin resistance gene and Stop sequences were flanked by loxP sequences and placed upstream of the sequence encoding Sharpin such that Sharpin was expressed under control of the *Rosa* promoter after Cre recombinase-mediated excision. TT2 (C57BL/6N × CBA) ES cells were transfected with the targeting vector. This strain

was used without crossing with Flp Deleter mice to detect targeted cells according to expression of the EGFP reporter. The PCR primers used to genotype the conditional Sharpin transgenic mice (accession no. CDB1201K: http://www2.clst.riken.jp/arg/mutant%20mice%20list.html) were as follows: Forward (wild-type allele), CCAGATGACTACCTATCCTC; Reverse (wild-type allele), GAGCTGCAGTGG AGTAGGCG; Forward (mutant allele) (f2), CTTGCCCTTCCTGCACTTTCATC; Reverse (mutant allele) (r4), GCAATATGGTGGAAAATAAC. Size of wild-type allele: 350 bp; size of floxed mutant allele: 300 bp.

**Cells and generation of CRISPR/Cas9-mediated KO cell lines**. HOIP-KO Jurkat cells and $Hoil-1l^{-/-}$ and $Sharpin^{-/-}$ MEFs were generated as described previously[46,47]. Gene-targeting KO B3Z, kindly provided by Takayuki Kanaseki (Sapporo Medical University, Japan), Jurkat, and $Sharpin^{-/-}$ MEF cells were generated using the CRISPR/CAS9 system. The CRISPR design tool (crispr.mit. edu) was used to select target guide sequences. The synthesized sgRNA oligos (see below) were phosphorylated, annealed, and inserted into the pair of BbsI sites in pSpCas9(BB)-2A-GFP (PX458, Addgene). The plasmids were transfected into cells by electroporation. After 24 h, GFP-expressing cells were purified using a FACSAria III cell sorter (BD Biosciences), according to the manufacturer's protocol, and single clones were obtained by limiting dilution. Disappearance of targeted proteins was determined by western blot analysis or flow cytometry analysis. The CRISPR sgRNA sequences (gene name, sgRNA number, and sequence) were as follows: human *Sharpin* sgRNA#1, GCGGTGGGCAGTCCTAGTCCG; human *Sharpin* sgRNA#2, GCCACGGTGGCACCTCGGACT; mouse *Sharpin* sgRNA#1, GTGGCAGTGCACGCGGCGGGTC; mouse *Sharpin* sgRNA#2, GACTGCCGCAC CGCCGGCGGG; mouse *Hoip* sgRNA#1, GTGGTCCGCTGCAACGCTCAT; mouse *Hoip* sgRNA#2, CACCGCAGCTGGACGCCGGACGCT; mouse *β2m* sgRNA, GTCACGCCACCCACCGGAGAA; mouse *Tap1* sgRNA#1, GTCCTAGG ACTAGGGGTCCGC; mouse *Tap1* sgRNA#2, GCCTAGGACTAGGGGTCCGCG; mouse *Tnfrsf1a* sgRNA#1, GTGTCACGGTGCCGTTGAAGC; mouse *Tnfrsf1a* sgRNA#2, GCACTCAGGTAGCGTTGGAAC.

**In vitro T cell-inducible cell death assay**. Primary T cells were isolated from secondary lymphoid tissues using anti-CD4, CD8α, or Thy1.2 MACS microbeads (Miltenyi Biotec). The T cells ($1–2 × 10^5$) were co-cultured for 10 h with MEFs (target cells; $5 × 10^4$) in a 96-well flat-bottom plate (200 μl/well). Floating T cells were removed by washing three times with medium. Survival of target cells was measured in a MTT assay (Dojindo, Japan), according to the manufacturer's protocol. For some experiments, z-VAD-fmk (20 μM, MBL) and/or Necrostatin-1 (50 μM, Sigma), a titrated neutralizing antibody against TNFα (XT3.11, Bio X cell), a rat IgG1 isotype control (HRPN, Bio X cell), or Fc-fused TNFRII (Etanercept, Pfizer) was added to the wells. For the siRNA knockdown experiment, target cells were plated in a 6-well plate ($1 × 10^5$/well). One day later, cells were treated with each siRNA (12.5 nM) diluted in Opti-MEM (Gibco). After 2 days, cells were harvested and used as target cells. The target genes and siRNA sequences were as follows: mouse *Fadd* siRNA#1, GGCACAUGUGCUGACUGCATT; mouse *Fadd* siRNA#2, GCAACGAUCUGAUGGAGCUTT; mouse *Ripk3* siRNA, CUCAAGUU CGGCCAAGUAUTT; mouse *Casp8* siRNA#1, GACAUAACCCAACUCCG AATT; and mouse *Casp8* siRNA#2, GUGAAUGGAACCUGGUAUATT.

**Staining of dead cells**. After exposure to T cell-inducible cell death assay immobilized in a collagen-coated plate, floating cells (including T cells) were removed by washing with PBS. The cells were then stained for 30 min at room temperature in the dark with TO-PRO3 (1 μM, ThermoFisher Scientific) and Hoechst33342 (5 μg/ml, Molecular Probes). After washing, photomicrographs of dead cells were acquired under a Keyence Fluorescence Microscope (BZ-9000) using CFI Plan Apo ×40 0.95/0.14 mm objective lenses (Nikon). The observed signals were processed using the BZ-II image analysis application (Keyence).

**Measurement of cell viability using iCelligence**. The real-time cell analyzer iCelligence system (ACEA Biosciences) was used as described[47]. Cell death inhibitors z-VAD-fmk (100 μM, MBL) and/or Necrostatin-1 (50 μM, Sigma) were added 1 h before administration of purified T cells.

**In vitro T cell stimulation**. Primary T cells were isolated from splenocytes using anti-Thy1.2 MACS microbeads (Miltenyi Biotec). Then, $1 × 10^6$ T cells were stimulated with a soluble anti-CD3ε antibody (1 μg/ml) or with a combination of an anti-CD3ε antibody (1 μl/ml) and a soluble anti-CD28 antibody (1 μg/ml).

**T cell hybridoma stimulation assay**. CD8$^+$ B3Z T cell hybridomas were stimulated by pOVA (SIINFEKL) presented on H2K$^b$ murine MHC class I molecules, as previously reported[48,49]. The pOVA was purchased from MBL. K$^b$-expressing L cells, kindly provided by Nilabh Shastri (University of California, USA), were used as APCs. The T cell hybridoma ($1 × 10^5$/well) was incubated for 24 h with the APCs ($1 × 10^5$/well) in the presence of titrated amounts of exogenous peptide. The culture supernatants were then added to IL-2-dependent HT-2 cells, kindly

provided from Eric Huseby (University of Massachusetts Medical School, USA), ($4 × 10^3$/well) and incubated for 24 h. Cytokine production by T cells was assessed by measuring proliferation of HT-2 cells in a MTT assay (Dojindo). All experiments were repeated three times, each with similar results.

**Antibodies used for flow cytometry analysis**. The following mouse antibodies were used: anti-CD25 (PC61, 102006), anti-IFNγ (XMG1.2, 505806), anti-CD103 (2E7, 121420), anti-GITR (YGITR765, 120205), anti-CD44 (IM7, 103008), anti-IL-4 (11B11, 504104), anti-CD4 (GK1.5, 100434), anti-CD25 (PC61, 102029), anti-IL-5 (TRFK5, 504303), anti-CD62L (MEL-14, 104412), anti-CD69 (H1.2F3, 104513), anti-Gr1 (RB6-8C5, 108411), anti-CD103 (2E7, 121403), anti-IL-17A (TC11-18H10.1, 506916), anti-GITR (DTA-1, 126311), anti-CD304 (Nrp1) (3E12, 145209), anti-CD3ε (145-2C11, 100334), anti-CD19 (6D5, 115519), anti-CD152 (CTLA4) (UC10-4B9, 106313), anti-B220 (RA3-6B2, 103224), streptavidin-PerCP (405213), anti-KLRG1 (2F1/KLRG1, 138409), anti-CD11b (M1/70, 101215), anti-Helios (22F6, 137216), anti-CD154 (CD40L) (MR1, 106505), anti-CD120a (TNFRI) (55R-286, 113003), anti-CD120b (TNFRII) (TR75-89, 113405), anti-CD262 (DR5, TRAIL-R2) (MD5-1, 119905), and anti-CD3ε (145-2C11, 100333) (all from BioLegend); anti-CD3ε (145-2C11, 553062), anti-CD11b (M1/70, 553310), anti-B220 (RA3-6B2, 11-0452-82), anti-CD5 (53-7.3, 553022), anti-CD25 (PC61, 552880), anti-CD8α (53-6.7, 552877), anti-CD19 (1D3, 562701), streptavidin-APC-Cy7 (554063), anti-CD69 (H1.2F3, 553237), anti-CD95 (Fas) (Jo2, 554258), and anti-SiglecF (E50-2440, 552126) (all from BD Biosciences); anti-TCRβ (H57-597, 11-5961-82), anti-CD25 (PC61.5, 12-0251-82), anti-Ki-67 (SolA15, 12-5698-80), anti-IFNγ (XMG1.2, 12-7311-81), anti-CD4 (RM4-5, 17-0042-82), anti-B220 (RA3-6B2, 17-0452-83), anti-IL-13 (eBio13A, 50-7133-80), anti-γδTCR (eBioGL3, 13-5711-81), streptavidin-APC (17-4317-82), anti-NK1.1 (PK136, 17-5941-81), anti-CD45RB (C363.16A, 12-0455-82), anti-Foxp3 (FJK-16s, 11-5773-82), anti-CD19 (eBio1D3, 11-0193-82), anti-TCRβ (H57-597, 12-5961-82), anti-IL-2 (JES6-5H4, 12-7021-81), streptavidin-PE (12-4317-87), anti-CD62L (MEL-14, 13-0621-82), anti-NK1.1 (PK136, 13-5941-82), anti-TNFα (MP6-XT22, 17-7321-81), anti-IL-10 (JES5-16E3, 17-7101-81), and anti-IFNγ (XMG1.2, 17-7311-81) (all from eBioscience); anti-CD127 (A7R34, 50-1271-U025) (TONBO Biosciences); and anti-p-ERK (D13.14.4E, 4344), anti-p-p65 (93H1, 4886), and anti-IκBα (L35A5, 7523) (all from Cell Signaling Technology). All antibodies used for flow cytometory analysis were diluted at 1:200 before staining.

**Western blot analysis**. Jurkat cells, thymocytes, MACS-purified T cells, or sorted (TCRβ$^+$CD4$^+$Foxp3$^+$CD25$^+$) Treg were lysed on ice in 1% Nonidet P-40/PBS/ protease inhibitor cocktail (Roche) and then clarified by centrifugation at 20,000×$g$ for 5 min at 4 °C. The purified lysate was used for SDS-PAGE and immunoblot analysis as described previously[7]. Chemiluminescent signals were detected using an ImageQuant LAS 4000 Image reader (Fujifilm). Antibodies against LUBAC components were described previously[6,7]. Anti-α-tubulin (DM1A, CLT9002, 1:5000 dilution) was purchased from Cedarlane Laboratories. Anti-β-actin (AC-74, A5316, 1:5000 dilution) was purchased from Sigma.

**Luciferase assay for measurement of NF-κB signaling**. The luciferase reporter plasmids pGL-NF-κB-Luc (10 μg) and pGL-TK-Luc (10 μg) (Promega) were electroporated (250 V, 950 μF) into Jurkat cells ($5 × 10^6$). After 48 h, cells were collected and stimulated with murine recombinant TNFα (20 ng/ml, R&D Systems) or anti-CD3ε/CD28 antibodies (1 μg/ml, BioLegend) for the indicated times. Next, the cells were lysed and ligand-dependent NF-κB activity was measured (in terms of luciferase activity) using the Dual-Luciferase Reporter or Bright-Glo luciferase assay system (Promega), according to the manufacturer's protocol. Luminescence was detected in a Lumat luminometer (Berthold).

**Lentiviral transduction of Jurkat cells**. Human wild-type or mutant *Hoip* cDNA was inserted into pCSII-EF-MCS-IRES2-Venus. Each plasmid was then transfected into 293 T cells along with pCMV-VSV-G-RSV-Rev and pMDLg/pRRE plasmids. After 24 h, the culture medium was replaced with fresh DMEM containing 10 μM forskolin and left for 72 h. Lentivirus in the culture supernatant was concentrated using a Lenti-X concentrator (Takara), and the lentiviral titer was determined by measuring Venus expression. Jurkat cells were infected with lentivirus (multiplicity of infection = 10) in the presence of polybrene (10 μg/ml). Infected Venus$^+$ Jurkat cells were enriched using a FACSAria III cell sorter (BD Biosciences), according to the manufacturer's protocol.

**Histologic analysis**. Mice were deeply anesthetized, and organs were removed and dissected (except for the lung). Each tissue was fixed immediately in 10% formalin solution buffered with PBS. Lung tissue was prefixed by perfusion with fixation solution via the trachea using a SURFLO ETFE I.V. Catheter (24G × 3/4″; TER-UMO) and then dissected and immersed in fixation solution for a further 48 h. The fixed organs were embedded in paraffin wax and sectioned. All sections were stained with H/E. Skin sections were stained with Masson's trichrome stain to visualize fibrosis in inflamed areas. Photomicrographs were acquired under an

Olympus BX51 upright microscope using UPlan Apo ×10/0.40, UPlan Apo ×20/0.70, and UPlan Fl ×40/0.75 objective lenses (Olympus).

**Immunofluorescence analysis of skin sections**. Formalin-fixed paraffin-embedded sections were deparaffinized, immersed in citrate buffer (10 mM citric acid, pH 6.0), and then boiled for 20–40 min in a microwave processor (MI-77; Azumayaika, Japan) for antigen retrieval. After sections were blocked for 30 min at room temperature with an anti-CD16/32 antibody (2 μg/ml) in 5% goat serum-containing blocking buffer (2% BSA, 0.1% Triton X-100, and 0.02% sodium azide in PBS), they were stained overnight at 4 °C with the following antibodies in blocking buffer: anti-CD3ε (HH3E, DIA-303; Dianova), anti-keratin14 (PRB-155P, Covance), anti-F4/80 (Cl:A3-1, MCA497GA; BIO-RAD), anti-myeloperoxidase (MPO) (RB-373-A1, ThermoFisher Scientific), anti-cleaved caspase-3 (Asp175, 9661; Cell Signaling Technology), and anti-phospho (S345)-MLKL (EPR9515(2), ab196436; Abcam). The stained sections were then incubated for 1 h at room temperature with a mixture of the following fluorescent dye-conjugated antibodies in blocking buffer: goat anti-rat IgG-AlexaFluor 647 (A-21247) and goat anti-rabbit IgG-AlexaFluor 488 (A-11034) (ThermoFisher Scientific). For preservation, labeled sections were mounted in ProLong Gold or ProLong Diamond Antifade Mountant (ThermoFisher Scientific) containing DAPI. Photomicrographs were acquired under a Keyence Fluorescence Microscope BZ-9000 using CFI Plan Apo ×40 0.95/0.14 mm objective lenses (Nikon) or an Olympus FLUOVIEW FV1000 confocal laser scanning microscope using a Plan Apo N ×60 /1.42 objective lens (Olympus). The observed signals were processed using the BZ-II image analysis application (Keyence).

**TUNEL staining**. Formalin-fixed, paraffin-embedded skin sections were used for TUNEL staining. After deparaffinization, sections were treated for 1 h at room temperature with Proteinase K (10 μg/ml in 10 mM Tris-HCl, pH 7.4). The TUNEL reaction was performed using the In Situ Cell Death Detection Kit, Fluorescein (Roche). For preservation, labeled sections were mounted in ProLong Gold Antifade Mountant (ThermoFisher Scientific) containing DAPI. Fluorescence images of apoptotic cells were acquired under a Keyence Fluorescence Microscope BZ-9000 using CFI Plan Apo ×40 0.95/0.14 mm objective lenses (Nikon). The observed signals were processed using the BZ-II image analysis application (Keyence).

**Cell isolation from tissues and staining for flow cytometry**. Thymocytes from 4- to 5-week-old mice, and splenocytes and lymph node cells from 4- to 10-week-old mice, were stained with a mixture of antibodies (see above). Cells infiltrating the liver of 8- to 10-week-old mice were isolated by mechanical dissociation using a 100 μm cell strainer (Falcon), followed by purification on a 33% Percoll solution (Sigma). Skin-resident cells were isolated separately from the dermis or epidermis. After shaving, the skin was immersed in 0.5 g/ml Dispase II (Wako) diluted in PBS and left at 4 °C overnight. The epidermal layer was then peeled from the dermis. Epidermal tissues were treated at 37 °C for 10 min with 0.05% DNase I (Roche) containing trypsin solution. Dermal tissues were incubated with 0.13 units/ml Liberase TM (Roche) and 100 μg/ml DNase I (Roche) diluted in plain RPMI1640 and then rotated at 37 °C for 40 min. The cell suspension obtained from each tissue was then filtered through 100 μm nylon filter mesh. For intracellular staining of transcription factors, including Foxp3, surface-labeled cells were fixed, permeabilized, and stained using the Foxp3/Transcription Factor Staining Buffer Set (eBioscience), according to the manufacturer's protocol. To examine the ability of T cells to produce cytokines, cells isolated from each organ were stimulated for 4 h at 37 °C with 50 ng/ml PMA and 750 ng/ml ionomycin in the presence of the protein transport inhibitors GolgiStop/GolgiPlug (1:1000; BD Biosciences). After surface staining, stimulated cells were fixed, permeabilized, and stained using the Fixation/Permeabilization Solution Kit (BD Biosciences), according to the manufacturer's protocol. For p-STAT5 staining, isolated cells were fixed for 1 h at room temperature with 4% formalin containing PBS and then treated with cold methanol for 1 h on ice before staining. For p-p65, IκBα, and p-ERK staining, stimulated cells were fixed for 10 min at 37 °C with Lyse/Fix buffer (BD Biosciences) and then washed with Perm/Wash buffer (eBioscience) before staining. To detect intracellular caspase activity in epidermal cells by flow cytometry, isolated cells were incubated at 37 °C for 30 min with PhiPhiLux-G1D2 (OncoImmunin, Inc.). All antibodies were used at 1:200, except for the anti-CD3ε antibody (used at 1:100). Flow cytometry analysis was performed using a FACSCanto II flow cytometer (BD Biosciences), according to the manufacturer's protocol. Data were analyzed using FlowJo software (Tomy Digital Biology).

**T cell-induced colitis and in vivo T_reg suppression assay**. Splenocytes and LN cells from 4–5-week-old B6 (CD45.1+) mice were depleted with anti-CD19 and anti-CD8α MACS microbeads (Miltenyi) to enrich CD4+ T cells. Next, pre-enriched cells were stained with a mixture of antibodies, including anti-CD25, anti-CD4, anti-CD3ε, and anti-CD45RB, followed by staining of dead cells with LIVE/DEAD Fixable Dead Cell Stains (Molecular Probes). Labeled cells were used for T cell sorting with a FACSAria III cell sorter (BD Biosciences) to purify CD45RBhiCD25-CD4+ naïve T cells. The sorted population was >98% pure. Foxp3/YFP+CD25+CD4+CD3ε+ (CD45.2+) Treg from control or Sharpin^fl/

fl*Foxp3^Cre* mice were prepared in a similar manner. Briefly, $2 \times 10^5$ purified naïve T cells, either alone or in combination with $1 \times 10^5$ purified control or Sharpin-KO Treg, were suspended in HBSS and injected intravenously into *Rag2^{-/-}* mice (five to six *Rag2^{-/-}* mice per group). Recipient mice were monitored for colitis progression by weighing two or three times per week. At 4 weeks post-transfer, recipient mice were sacrificed for further experiments.

**Quantitative PCR analysis of skin**. After removing hair with an electric shaver, part of the dorsal skin was dissected. Pre-cleaned RNAs were extracted using Sepasol-RNA I Super G (Nacalai Tesque, Japan), according to the manufacturer's protocol, and then subjected to column-based purification using an RNeasy Mini Kit (Qiagen). Skin cDNA was obtained by reverse transcription of RNA using a High-Capacity RNA-to-cDNA Kit (Applied Biosystems), followed by qPCR analysis using SYBR Green PCR Master Mix (Applied Biosystems) and a ViiA7 Real-time PCR system (Applied Biosystems). The qPCR amplification conditions were as follows: 95 °C for 10 min, followed by 40 cycles of 95 °C for 15 s and 60 °C for 1 min. Expression of each gene was normalized against that of hypoxanthine-guanine phosphoribosyl transferase (HPRT). The following primers were used: *mouse HPRT*, Forward, GATTAGCGATGATGAACCAGGTT, and Reverse, CCTCCCATCTCCTTCATGACA; *mouse IL-6*, Forward, TACCACTTCACAAG TCGGAGGC, and Reverse, CTGCAAGTGCATCATCGTTGTTC; *mouse TNFα*, Forward, GGTGCCTATGTCTCAGCCTCTT, and Reverse, GCCATAGAAC TGATGAGAGGGAG; *mouse IL-1β*, Forward, TGGACCTTCCAGGATGAGG ACA, and Reverse, GTTCATCTCGGAGCCTGTAGTG; and *mouse IL-4*, Forward, ATCATCGGCATTTTGAACGAGGTC, and Reverse, ACCTTGGAAGCCCTA CAGACGA.

**Scoring for dermatitis**. The disease scoring system was described previously[50]. In brief, the skin was divided into four regions (head; neck and/or back; front limbs and/or ventral area; and hind limbs and/or rump area). Each inflamed region scores 1 point, except for the head, which scores 2 points. Additionally, the severity of each affected area was graded as follows: 0 = no disease; 1 = hair loss and mild scaling; 2 = extensive scaling and erosion; 3 = ulceration. The final score for each individual mouse was the sum of the scores for each affected region plus the severity grade (maximum score, 17 points). All mice were scored two or three times per week during the experiment.

**B cell or myeloid cell depletion**. To deplete B cells or myeloid cells, 4-week-old mice received an intraperitoneal injection of both anti-CD19 (1D3) and anti-B220 (RA3.3A1/6.1), or anti-Gr1 (RB6-8C5), or Isotype control (LTF2) antibodies; these injections were given five times per week.

**In vivo TNFα neutralization**. A total of 200 μg of an anti-TNFα Ab (clone XT3.11) or isotype control IgG (clone HRPN) was intraperitoneally administered to 3-week-old mice. The mice received one injection every 3 days, and that was repeated eight times.

**Microarray analysis**. Total RNA was extracted from FACS-sorted Treg using an RNeasy Mini Kit (Qiagen), according to the manufacturer's instructions. The purity of the RNA was checked using a NanoDrop 2000 spectrophotometer (ThermoFisher Scientific) and a 2100 Bioanalyzer (Agilent Technologies). Biotinylated cRNA was synthesized from 250 ng of total RNA using the GeneChip 3′ IVT PLUS Reagent Kit (Affymetrix), according to the manufacturer's instructions. The yield of biotinylated cRNA was checked using the NanoDrop 2000 spectrophotometer. Following fragmentation, 15 μg of cRNA was hybridized to a Gene-Chip Mouse Genome 430 2.0 Array (Affymetrix) for 16 h at 45 °C. Arrays were washed and stained in a GeneChip Fluidics Station 450 (Affymetrix). Arrays were scanned using GeneChip Scanner 3000 7G. Single Array Analysis was calculated using Microarray Suite version 5.0 (MAS5.0), with the Affymetrix default setting and global scaling as the normalization method. The trimmed mean target intensity of each array was arbitrarily set to 500.

**Statistical analysis**. Statistical analysis was performed using Prism 5.0c (Graph-Pad software). Statistical significance was determined using a two-tailed Mann–Whitney U-test, a paired *t*-test, or the Kruskal-Wallis test, followed by pairwise comparisons with Bonferroni-corrected Dunn's test. A *p*-value of 0.05 was deemed significant.

**Reporting summary**. Further information on research design is available in the Nature Research Reporting Summary linked to this article.

## Data availability

The Gene Expression Omnibus accession number for the transcriptional data reported in this study is GSE108793. The authors declare that all other data of this study are available within the paper or upon reasonable request. The source data are provided in a Source Data file.

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

## Acknowledgements

We thank the Anatomic Pathology Center of the Graduate School of Medicine, Kyoto University, for preparing tissue sections. We also thank Sho Hanakawa, Kenji Kabashima, Miho Sekai, Yoko Hamazaki, Nagahiro Minato (Kyoto University, Japan), and Alexander Rudensky (Memorial Sloan Kettering Cancer Center, USA) for help with generating mutant mice. We would like to thank Eric Huseby (University of Massachusetts Medical School, MA, USA), Nilabh Shastri (University of California, USA), Yoshinori Fukui (Kyushu University, Japan), and Takayuki Kanaseki (Sapporo Medical University, Japan) for providing the cell lines. Finally, we thank all members of Kazuhiro Iwai's lab for technical support and discussion. This work was supported by JSPS KAKENHI Grant Numbers 24112002, 25253019, 26670154, 17H06174, and 18H05499 (to K.I.) and 16K19106 (to K.S.).

## Author contributions

K.S. and K.I. designed the experiments, interpreted the results, and wrote the paper. K.S. performed the experiments. A.H. and T.N. performed flow cytometry analysis. Y.S. and K.I. generated conditional HOIP and Sharpin knockout mice. H.K. generated conditional Sharpin transgenic mice.

## Additional information

**Competing interests:** The authors declare no competing interests.

