## [Peer Review File · Nature Communications]

Reviewers' comments:

Reviewer #1 (Skin inflammation)(Remarks to the Author):

This manuscript utilizes primarily mouse models to investigate the contribution of the LUBAC to autoinflammation and in particular to link LUBAC in Tregs to T cell mediated autoinflammatory disease with a primary focus on dermatitis.

The overall concept is interesting and novel. The authors should be commended on what clearly was a significant amount of work. My major criticisms are two-fold. First, the manuscript is difficult to follow both due to limitations in writing (sentence structure in particular and word choice) and to organization. For example, I had to reread several sentences to really appreciate the authors intent and in some cases was unable to understand, and I had to revisit repeatedly figures and figure legends and then text to follow along.

Second, there is a lot of extraneous data that distracts from the main purpose of the study and there are several similar though unique mouse models utilized in each figure that seem to be haphazardly compared (or not) to each other making the rationale for choice of model difficult to understand. Perhaps excluding some data, simplifying the text in the results section, and reorganizing/reordering some of the figures may help the manuscript to flow more clearly.

Additional points:

Some of the figures are quite small to the point they are difficult to see (examples figure 2e, blots in supplemental figure 2c, Fig 3b way too small).

Please do not use similar colors for different groups on the same graph (ie red and light red; blue and light blue) its hard to distinguish.

Stats – it appears that for the most part the authors perform Mann Whitney test –I am not a statistician but I think of the Mann Whitney as the nonparametric version of a Ttest – in many cases though it seems that multiple tests are being performed in these studies from within the same experiment (Examples Figure 2g, sup. Fig 1B, sup. Figure 2A and C). If so, these would require a non-parametric version of an ANOVA (?Kruskal Wallis) and adjusted p values for significance to take into account multiple testing (Bonferroni correction?). Please have a statistician review your statistical analyses.

Specific comments:

- Simplify/shorten the intro. In particular, do not summarize your findings in such great detail in the intro – that's for the conclusion

Results:

- Perhaps move supplemental figure 1 and 2 to later?? – rather start with Figure 1 in the results section.

- Figure 1. Say how you are defining your "Treg" vs "Tconv" I assume CD4+ CD25+ Foxp3+ for Treg. I recommend moving the colitis (g) to supplemental data. Instead focus on the skin sharpinflux/foxp3 cre model with emphasis on the skin.

- Since you've shown histology from the lung I'm left wondering whether all the other organs are normal or inflamed; if all others are normal you can write in text, no need to show normal histology. You do this for other mouse models later in the paper.

- Can you make the histology bigger so we can see it? And are you sure your diseased mouse skin H&E is the same magnification as the healthy? If so, you have remarkable thickening of the epidermis and dermis, as well as impressive cellular infiltrates – I can't see well enough to tell if centered around a hair follicle and/or other adnexal structures and/or perivascular. I ask b/c I can make out the cells on the diseased H&E epidermis but not the healthy suggesting a different magnification. Lastly, the authors repeatedly make comparisons of this and other mouse models in their manuscript to cpdm

like skin inflammation – can you show cpdm like skin inflammation clinically and histologically here so we can see the comparison in this figure --- or include in the text where the images can be found (I found it later in a different figure but you had already made several comparisons to it before you showed it).

- Next show Supp. Fig 1 sharpin floxK5cre which also have skin disease but seem to lose their foxp3 expression on treg.

o How do you know that Treg are losing their foxp3 expression as opposed to having increased numbers/percentages of Tcon (since presumably you're defining Treg on the Foxp3 expression, and since activated Tconv express Cd25)

o For Supp Fig 1D – show the skin H&E so we can compare it to the H&E in the Foxp3cre mice from Figure 1 (and to the cpdm skin inflammation, see above).

o Since the keratinocytes are not intact per your text it would mean the epidermal barrier may be compromised and therefore it is possible microbial organisms have gained entrance to deeper skin levels inducing an inflammatory response – for example your CD44hiCD62Llo cells could represent activated Tconv.

o How do you reconcile that you have keratinocyte death yet the epidermis is thickened with hyperkeratosis (like in you Figure 1i) rather than denuded or blistered which one would expect if significant keratinocyte cell death

o For (g), what about later time point like in E and F which indicated that Treg markers are skewed over time?

- Figure 2.

o for e and f are you only including the sick mice for the sharpin/lck model or all mice? I would separate the data for sick mice from the nonstick sharpin/lck – might be very interesting.

o CD44hi + CD62L alone are not great markers alone for defining T cell populations – in particular Cd44hiCD62Llo –how are you distinguishing recently activated effector cells from effector memory circulating through?

o what about looking at actual activation markers on CD4 Tconv and CD8 T cells

o the authors utilize the colitis model as a marker of T cell mediated inflammation however, in truth all they show in multiple figures is weight loss. Weight loss alone is insufficient to demonstrate T cell mediated inflammation – histology demonstrating inflammation + actual immunologic assays or at least T cells stains comparing to controls would be necessary. Essentially the data presented in figure 2 does not convincingly show that T cells from Sharpin^{fl/fl}Lck^{cre} mice are less capable of function

o can you clarify what you mean by “these data suggest that cpdm-like skin inflammation observed in sharpin^{fl/fl} foxp3cre mice is caused by autoimmune prone T cell traits??? What traits are you referring to?

- Fig 3.

o Please improve the images in 3b -I cannot see the TUNEL staining (or even make out the cells at all). Is the dotted line your dermal-epidermal junction?

o Why look at TNFa, IL1b and IL6 in cpdm mice and R26 variants but not in the sharpin^{fl/fl} Foxp3cre mice? How do they compare (based on the clinical pics it seems like the cpdm inflammation is much more mild than when sharpin is only missing from Foxp3+ cells).

o To this point, can you show the TUNEL staining in the other mouse models particularly the sharpin^{fl/fl} Foxp3cre? [may want to reorganize results altogether as mentioned above).

- Fig 4.

o The title compares Sharpin^{fl/fl} Foxp3cre to sharpin^{fl/fl} K5cre but (a) shows the K5cres and cpdms why not the foxp3cres (which are down in (g) – likewise why do the MT stain on the Foxp3cres but not the cpdm and K5cres?

o Same for b, c, d, and h

- Fig 5.

o (a and b) can you quantify the number of casp 3 and pMLKL in K5cre vs Foxp3cre vs ctrls for example per LPF?

o Why do you think dermal fibroblasts are dying in the K5cre mice?

I may have missed this but what happens if you delete T cells altogether in sharpinfl/flK5 mice? Presumably the mice should be normal if you think T cells are promoting autoinflammation. If you treat with a neutralizing TNF α antibody in vivo, can you prevent or treat disease?

Lastly, out of pure interest – have you looked at the mucosa in any of these mouse models? There is a disease termed SJS/TEN (Stevens Johnson Syndrome/Toxic Epidermal Necrolysis) that is typically attributed to drugs but has some interesting potential overlaps in regards to t cell mediated keratinocyte death that might be a truer clinical corollary to your findings rather than generic dermatitis.

Reviewer #2 (Ubiquitination, NFkB signalling)(Remarks to the Author):

In this manuscript, authors present a series of studies suggesting that Sharpin deficiency in Tregs causes a cpdm-like skin inflammation, which is t cell-dependent dermatitis. Recovered expression of Sharpin in t lineages rescues cpdm mice from skin inflammation. Skin-oriented activated T cells promote apoptotic and necroptotic keratinocyte death and induce skin autoinflammation. Further, authors found T cell-expressed TNF- α can induce cell death to regulate skin inflammation. These functional studies provide new insight into the function of Sharpin and LUBAC in the regulation of inflammation. However, the work is somewhat descriptive and could be strengthened by including more in depth mechanistic studies.

Specific comments:

1. How Sharpin regulates Treg function was not sufficiently investigated. The authors used Sharpin-deficient Jurkat cells to show a reduction in the stability of LUBAC and activation of NF- κ B (sFig. 2a). However, it is unclear whether and how Sharpin regulates TCR signaling in Treg cells. To confirm the conclusion that ablating Sharpin in Tregs causes skin inflammation through partial impairment of TCR-mediated signaling, it is important to analyze the stability of LUBAC and signaling (such as NF- κ B activation) in Sharpin-deficient Treg cells.
2. Fig.1h-K showed Sharpinfl/flFoxp3Cre mice had lymphadenopathy with increased numbers of CD3+ T cells, B220+ B cells, and CD11b+ myeloid cells in pLNs. How about the immune cell infiltration in the skin in Treg-specific Sharpin-deficient mice.
3. Fig.2 and 3 showed skin inflammation observed in Treg-specific Sharpin-deficient mice is dependent on T cells. To rule out the role of other immune cells, the authors should deplete B220+ B cells and CD11b+ myeloid cells in Sharpin fl/flFoxp3Cre mice to analyze the skin inflammation.
4. SFig.1 showed that keratinocyte-specific Sharpin-deficient mice developed cpdm-like dermatitis. Does Sharpin regulate apoptotic and necroptotic cell death in keratinocyte cells? Keratinocyte-specific Sharpin-deficient cells showed a reduction in Foxp3 expression and an activated phenotype. What is the potential mechanism by which Sharpin-deficient keratinocytes regulate Treg phenotype?
5. In Fig.2, Sharpin fl/flLCKCre mice showed the reduced incidence of dermatitis and no changes in the population and phenotype of peripheral T cell lineages, compared with the Sharpin fl/flFoxp3Cre . The data suggest sharpin may be required for activation of CD4 or CD8 T cells. The authors should analyze the activation of CD4 and CD8 T cells from Sharpin fl/flLCKCre and wildtype control mice.
6. Fig4d Sharpinfl/flK5Cre mice showed the tendency of reduced self-reactive CD4+ CD69+ T cells, The authors should repeat it by quantifying both the % and absolute cell number and by using more mice. It will also be interesting to perform parallel analysis using the Sharpinfl/flFoxp3Cre mice.
7. Fig.5 showed that apoptotic and necroptotic keratinocyte death were increased in both Sharpinfl/flFoxp3Cre and Sharpinfl/flK5Cre mice. What is the mechanism by which keratinocytes cell death induce skin autoinflammation and recruit CD11b+ myeloid cell infiltration?

8. Fig.6i showed activated T cells increased the expression of TNF- α , which can induce cell death. To study the increased cell death and skin inflammation, the authors should analyze the TNF- α expression in T cells from Sharpinfl/flFoxp3Cre mice.

9. Fig.6j showed that the skin autoinflammation was attenuated in Sharpinfl/flFoxp3Cre TNFa $^{-/-}$ mice and Sharpinfl/flFoxp3Cre RIPK3 $^{-/-}$ mice. The authors should analyze the apoptosis and necroptosis of keratinocytes and dermal fibroblasts, showing that cell death induced skin autoinflammation.

Reviewer #3 (Treg biology, autoimmunity)(Remarks to the Author):

The manuscript by Sasaki and colleagues studied the importance of LUBAC in Treg cells. The authors generated various conditional knockout mouse lines in which Sharpin or Hoip is specifically deleted in Tregs, T cells or Keratinocytes. The authors showed that knockout Sharpin in Tregs induced some similar skin phenotypes as those observed in global KO and keratinocyte specific KO of Sharpin, suggesting an important role of Tregs in the development of dermatitis. Mechanistically, reduced Treg function resulted in T cell mediated keratinocyte death through mainly the TNF- α pathway. The studies were generally well performed and data were comprehensive. Several issues however need to be further addressed.

Major points:

- 1) Some the terminologies were used in a very confusing way throughout the study. For example, autoimmunity and autoinflammation are very difficult to distinguish and probably overstated in the study, since the authors were never provided any concrete data to support any of the phenotypes from various mouse lines fitting the restrictive definitions of these two terminologies, such as autoantigen/autoreactive T cells, or overexpression of either IL-1 or TNFa from innate immune cells. Therefore, the authors should remove (or minimize) the use them especially in a comparative and contrast way throughout the paper.
- 2) Although the authors generated many different mouse lines, the paper was a little descriptive. It would be helpful, if the authors could provide some mechanistic insights on how Sharpin/LUBAC impacted the Treg function and development.

Minor points

- 3) In Figure 6, the authors demonstrated an important role of TNFa pathway in T cell mediated keratinocyte death. But there is no data distinguished the role of soluble or membrane bound TNF. So in line 311, the authors should not use "membrane-bound TNFa".
- 4) The authors need to further edit the language throughout the paper.

Reviewers' comments

Point-by-point responses:

Reviewer #1

This manuscript utilizes primarily mouse models to investigate the contribution of the LUBAC to autoinflammation and in particular to link LUBAC in Tregs to T cell mediated autoinflammatory disease with a primary focus on dermatitis. The overall concept is interesting and novel. The authors should be commended on what clearly was a significant amount of work.

Major criticisms are two-fold. First, the manuscript is difficult to follow both due to limitations in writing (sentence structure in particular and word choice) and to organization. For example, I had to reread several sentences to really appreciate the authors intent and in some cases was unable to understand, and I had to revisit repeatedly figures and figure legends and then text to follow along. Second, there is a lot of extraneous data that distracts from the main purpose of the study and there are several similar though unique mouse models utilized in each figure that seem to be haphazardly compared (or not) to each other making the rationale for choice of model difficult to understand. Perhaps excluding some data, simplifying the text in the results section, and reorganizing/reordering some of the figures may help the manuscript to flow more clearly.

We very much appreciate the reviewer's helpful comments. As requested, we rearranged the main and supplemental figures to allow readers to follow more easily. Also, the text has been simplified accordingly. As for the data, we admit that the manuscript contains a large amount. However, we believe that all the data are necessary to illustrate the main concepts; therefore, we feel that we cannot exclude any data from the revised version. Our point-by-point responses to each of the reviewers' comments are outlined below.

Additional points:

Some of the figures are quite small to the point they are difficult to see (examples figure 2e, blots in supplemental figure 2c, Fig 3b way too small).

We have increased the size of the figures.

Please do not use similar colors for different groups on the same graph (ie red and light red; blue and light blue) its hard to distinguish.

We agree. We have revised the colors accordingly. Please see new Figures S1D, 3 (b, e–h), and S4B.

Stats – it appears that for the most part the authors perform Mann Whitney test – I am not a statistician but I think of the Mann Whitney as the nonparametric version of a Ttest – in many cases though it seems that multiple tests are being performed in these studies from within the same experiment (Examples Figure 2g, sup. Fig 1B, sup. Figure 2A and C). If so, these would require a non-parametric version of an ANOVA (?Kruskal Wallis) and adjusted p values for significance to take into account multiple testing (Bonferroni correction?). Please have a statistician review your statistical analyses.

We appreciate the helpful comment. As suggested, we used a Kruskal-Wallis test (non-parametric multiple comparison) with a Bonferroni correction (α value = 0.05) to compare all paired data in the new Figures S1 (D, G, J), 2h, 3 (b, e–h), S4 (B, G), 6(b–d, g, h), 7(c, d), and S5H.

Specific comments:

- Simplify/shorten the intro. In particular, do not summarize your findings in such great detail in the intro – that's for the conclusion

Thank you, we have simplified the “Introduction” section accordingly.

Results:

- Perhaps move supplemental figure 1 and 2 to later?? – rather start with Figure 1 in the results section.

Thank you. We now start the results section with Figure 1 (from the original manuscript) and have moved the original Figs. S1 to Fig. S4 to later in the revised version. We have revised the introduction accordingly. These changes do not alter the main conclusions described in the original manuscript. We hope that these changes make the manuscript easier to follow.

- Figure 1. Say how you are defining your “Treg” vs “Tconv” I assume CD4⁺ CD25⁺ Foxp3⁺ for Treg.

We defined T_{reg} or Tconv cells as CD4⁺Foxp3⁺T cells (the majority was CD25⁺Helios⁺Nrp1⁺ thymic T_{regs}, as seen in Fig. 1b and S1I) and CD4⁺Foxp3⁺T cells, respectively.

I recommend moving the colitis (g) to supplemental data. Instead focus on the skin sharpin^{fl/fl}/foxp3 cre model with emphasis on the skin.

As suggested by the reviewer, we have moved the colitis data to the new Fig. S1J; we now show some pictures of skin lesions in *Sharpin^{fl/fl}Foxp3^{Cre}* and *cpdm* mice at different magnifications (see Fig. 1i of the revised manuscript).

- Since you've shown histology from the lung I'm left wondering whether all the other organs are normal or inflamed; if all others are normal you can write in text, no need to show normal histology. You do this for other mouse models later in the

paper.

We observed some inflammatory changes in the lung, but the other organs appeared to have no overt abnormalities. We have moved the lung histology images to the new Fig. S1K and described the histological changes in the organs in the revised manuscript text. This was also done for the other mouse models.

- Can you make the histology bigger so we can see it? And are you sure your diseased mouse skin H&E is the same magnification as the healthy? If so, you have remarkable thickening of the epidermis and dermis, as well as impressive cellular infiltrates – I can't see well enough to tell if centered around a hair follicle and/or other adnexal structures and/or perivascular. I ask b/c I can make out the cells on the diseased H&E epidermis but not the healthy suggesting a different magnification. Lastly, the authors repeatedly make comparisons of this and other mouse models in their manuscript to cpdm like skin inflammation – can you show cpdm like skin inflammation clinically and histologically here so we can see the comparison in this figure --- or include in the text where the images can be found (I found it later in a different figure but you had already made several comparisons to it before you showed it).

The magnification used for the skin histology images of control and diseased mice is identical. To evaluate differences between control and diseased mice more easily, we also show higher magnification images of *Sharpin^{fl/fl}Foxp3^{Cre}* mice (new Fig. 1i, bottom). Histological data for *cpdm* skin are also presented in this figure, as suggested. We also described the comparison in the revised text.

- Next show Supp. Fig 1 sharpin flox K5cre which also have skin disease but seem to lose their foxp3 expression on treg.

o How do you know that Treg are losing their foxp3 expression as opposed to having increased numbers/percentages of Tcon (since presumably you're defining Treg on the Foxp3 expression, and since activated Tconv express Cd25)

We agree with the reviewer regarding CD25 expression by activated Tconv as well as T_{regs}. Since we do not have a fate mapping system based on a lineage-tracing mouse model, we examined the MFI of Foxp3 within T_{regs} (Foxp3-expressing CD4⁺T cells) and found that it was clearly falling in *Sharpin^{fl/fl}K5^{Cre}* as the disease progressed (see new Fig. S4E). Therefore, we concluded that T_{regs} are losing Foxp3 expression. Also, we used young mice (around 4 weeks old) for comparative phenotypic analyses of Tconv and T_{regs} (see new Fig. S4G) because they harbor T_{regs} expressing a sufficient amount of Foxp3, thereby allowing separation of the two T cell subsets.

o For Supp Fig 1D – show the skin H&E so we can compare it to the H&E in the Foxp3cre mice from Figure 1 (and to the cpdm skin inflammation, see above).

Due to restrictions on the amount of data, we are unable to show all of the H&E images from different mouse lines within the same figure. *Cpdm* and *Sharpin^{fl/fl}K5^{Cre}* are shown in new Figs 1i and S4D, respectively.

o Since the keratinocytes are not intact per your text it would mean the epidermal barrier may be compromised and therefore it is possible microbial organisms have gained entrance to deeper skin levels inducing an inflammatory response – for example your CD44hiCD62Llo cells could represent activated Tconv.

Epidermal keratinocytes in *Sharpin^{fl/fl}Foxp3^{Cre}* and *Sharpin^{fl/fl}Lck^{Cre}* are genetically intact. While skin inflammation was overt in the former, it was almost cured by additional loss of Sharpin from effector T cells in the latter, which indicates that the disease was definitely T cell-dependent. Thus,

involvement of invading microbial organisms in the onset of dermatitis is unlikely in these mice, although we cannot completely rule out the possible involvement of microorganisms in inflammatory responses after the skin barrier is disrupted. On the other hand, as the reviewer points out, epidermal integrity is disrupted in *Shardin^{fl/fl}K5^{Cre}* and *cpdm* mice. Although no activation of conventional T cells was observed (new Fig. 4f and S4G), innate immune responses and a specific type of cell death (pyroptosis), triggered by sensing of exogenous organisms and by inflammasome activation, may exacerbate skin inflammation (Ref. Douglas et al., JI, 2015, 195(5): 2365-73).

o How do you reconcile that you have keratinocyte death yet the epidermis is thickened with hyperkeratosis (like in you Figure 1i) rather than denuded or blistered which one would expect if significant keratinocyte cell death

It is hard to reconcile this at this moment in time. As far as we know, this question has not been resolved, although it has been asked for a long time. Our own findings and those of others support the fact that genetic defects resulting in suppression of NF- κ B signaling pathway induce skin inflammation along with epidermal hyperkeratosis and parakeratosis (Yenkel et al., JI, 2015, 194(6): 2472-2476). Thus, appropriate activation of NF- κ B in keratinocytes under steady state conditions would be required for resistance to cell death, maintenance of immune homeostasis, and normal cornification. As shown in the new Fig. 5A and B, some keratinocytes in the lesions are dying; however, most are viable and the epidermis is thickened, even in cases where skin inflammation appears to be very severe. We hypothesize that spontaneous keratinocyte cell death acts as a trigger that activates innate immune responses, leading to autoinflammatory dermatitis and, at the same time, induction of keratinocyte proliferation through a host of proinflammatory cytokines and factors secreted upon keratinocyte cell death.

o For (g), what about later time point like in E and F which indicated that Treg markers are skewed over time?

The phenotypic skewing of T_{reg} markers in *cpdm* mice is retained up until at least 14 weeks after birth, as shown in the new Fig. S4G.

- Figure 2.

o for e and f are you only including the sick mice for the sharpin/lck model or all mice? I would separate the data for sick mice from the nonstick sharpin/lck – might be very interesting.

Non-sick mice were used for evaluation of T lineage cells because we wanted to investigate any predisposition toward *Sharpin^{fl/fl}Foxp3^{Cre}*-like dermatitis in healthy *Sharpin^{fl/fl}Lck^{Cre}*. A few of the sick mice analyzed exhibited similar symptoms, including dermatitis, which were not different from those observed in *Sharpin^{fl/fl}Foxp3^{Cre}* (see below and please refer to the new Fig. S1l, 2c, 4c, 4f, and 7c). Thus, we do not believe that immunological phenotypes in sick *Sharpin^{fl/fl}Lck^{Cre}* mice help to answer our question because their immune cells are already exposed to an inflammatory environment. In addition, because of difficulties with breeding, we decided not to include the data in the manuscript.

“(A) Number of the indicated immune cells in pLNs from nonsick and Sick *Sharpin^{fl/fl}Lck^{Cre}* mice. (B) Percentage of CD69⁺ or CD44^{hi}CD62L^{lo} activated conventional T cells in pLNs (upper: CD4⁺T cells; lower: CD8⁺T cells). Small horizontal lines indicate the mean (\pm s.e.m.)”

o CD44^{hi} + CD62L alone are not great markers alone for defining T cell populations – in particular Cd44^{hi}CD62L^{lo} –how are you distinguishing recently activated effector cells from effector memory circulating through? what about looking at actual activation markers on CD4 Tconv and CD8 T cells

We examined expression of CD69 as a marker of T cell activation (see new Fig 2d and 4f). During our investigation of conventional T cells isolated from 3-month-old non-sick *Sharpin^{fl/fl}Lck^{Cre}*, we realized that they harbored an increased number of CD44^{hi} cells within the CD8⁺T cell population, but not in the CD4⁺T cell population. Consistent with this, we found that a small fraction of CD8⁺T cells was skewed to express high levels of CD69, an activation marker of T cells (see new Fig. 2c and d). Thank you for the helpful

and constructive comment.

o the authors utilize the colitis model as a marker of T cell mediated inflammation however, in truth all they show in multiple figures is weight loss. Weight loss alone is insufficient to demonstrate T cell mediated inflammation – histology demonstrating inflammation + actual immunologic assays or at least T cells stains comparing to controls would be necessary. Essentially the data presented in figure 2 does not convincingly show that T cells from Sharpin^{fl/fl}Lck^{cre} mice are less capable of function

As seen in Fig. 2h (weight loss data), Sharpin^{-/-} T cells (Sharpin^{fl/fl}Lck^{cre} mice) induced less severe T cell-dependent colitis than wild-type T cells. However, we could not see significant differences in inflammation-induced shortening of the colon or in histological analysis of T cells in the colon at 8 weeks post-transfer (Fig. 2i). Therefore, to evaluate whether loss of Sharpin attenuated the functions of effector T cells, we examined the immunological phenotypes of transferred naïve and Sharpin^{-/-} CD4⁺T cells in colon-draining mesenteric LNs; we found that the activated T cell subset (CD25⁺) within the Sharpin^{-/-} T cell population was smaller (see new Fig. 2j). Since we demonstrated that Sharpin is required for TCR-mediated T cell activation (see new Fig. 1 and 2e, g), our results clearly indicate that the reduced proinflammatory efficacy of Sharpin^{-/-} T cells underlies amelioration of colitis.

o can you clarify what you mean by “these data suggest that cpdm-like skin inflammation observed in sharpinfl/fl foxp3cre mice is caused by autoimmune prone T cell traits???” What traits are you referring to?

The skin inflammation observed in Sharpin^{fl/fl}Foxp3^{Cre} mice was indistinguishable from that in cpdm mice. Therefore, we referred to skin inflammation in Sharpin^{fl/fl}Foxp3^{Cre} mice as “cpdm-like”. We showed that T

cells with an activated phenotype trigger skin inflammation in *Sharpin^{fl/fl}Foxp3^{Cre}* mice by inducing keratinocyte cell death. Thus, our results imply a crucial role for T cell-mediated cytotoxicity in *cpdm*-like chronic skin disruption. However, to avoid confusion, we have decided to remove the sentence from the revised manuscript.

- Fig 3.

o Please improve the images in 3b -I cannot see the TUNEL staining (or even make out the cells at all). Is the dotted line your dermal-epidermal junction?

We agree. We have performed the TUNEL staining again (see new Fig. 3d). Since DAPI nuclear staining aids recognition of dermal and epidermal areas in the new figures, we have removed the dotted line from the revised figure.

o Why look at TNF α , IL1 β and IL6 in cpdm mice and R26 variants but not in the sharpinfl/fl Foxp3cre mice? How do they compare (based on the clinical pics it seems like the cpdm inflammation is much more mild than when sharpin is only missing from Foxp3+ cells).

***Rosa26 Sharpin-Tg cpdm* mice were established and used to examine whether T cell-mediated inflammation is involved in development of dermatitis in *cpdm* mice. Thus, we compared *Sharpin-Tg cpdm* mice with *cpdm* mice and observed amelioration of skin inflammation. With respect to expression of TNF α , IL-1 β , and IL-6 genes in the skin, *Sharpin^{fl/fl}Foxp3^{cre}* and *cpdm* mice exhibited skin inflammation of comparable severity.**

o To this point, can you show the TUNEL staining in the other mouse models particularly the sharpinfl/fl Foxp3cre? [may want to reorganize results altogether as mentioned above].

As suggested, we repeated the TUNEL staining of skin in another mouse model; the results are shown in the new Fig. 5a.

- Fig 4.

o The title compares Sharpin^{fl/fl} Foxp3^{cre} to sharpin^{fl/fl} K5^{cre} but (a) shows the K5^{cre}s and cpdms why not the foxp3^{cre}s (which are down in (g) – likewise why do the MT stain on the Foxp3^{cre}s but not the cpdm and K5^{cre}s? o Same for b, c, d, and h

We re-organized the figure and compared Sharpin^{fl/fl}Foxp3^{Cre} mice with Sharpin^{fl/fl}K5^{Cre} mice (see new Fig. 4).

- Fig 5.

o (a and b) can you quantify the number of casp 3 and pMLKL in K5^{cre} vs Foxp3^{cre} vs ctrls for example per LPF?

We counted these cells; the results are shown in the graph (see new Fig. 5b).

o Why do you think dermal fibroblasts are dying in the K5^{cre} mice?

We thank the reviewer for the thoughtful comment. We found that TUNEL-positive cells were relatively brighter in the dermis, indicating that dying cells were not restricted to epidermal tissue. However, we have no convincing results to indicate that the TUNEL-positive cells are dermal fibroblasts. We have removed the sentence from the revised manuscript.

I may have missed this but what happens if you delete T cells altogether in sharpin^{fl/fl}/fIK5 mice? Presumably the mice should be normal if you think T cells are promoting autoinflammation.

Accumulated evidence, including a previous report showing that cpdm

Rag1^{-/-} mice, which lack T cells, develop skin inflammation, supports the idea that T cell-mediated autoinflammation might not play a pivotal role in the pathogenesis of *Sharpin*^{fl/fl}*K5*^{cre} dermatitis. Furthermore, we established *TCRα*^{-/-}*Sharpin*^{fl/fl}*K5*^{cre} mice and found that they also develop skin inflammation, as was the case for *Sharpin*^{fl/fl}*K5*^{cre} mice. However, the data are not presented in the main and supplemental figures of the revised manuscript because they are not related directly to the main aims of the study.

A

Sharpin^{fl/fl}*K5*^{Cre}*TCRα*^{-/-}

6w

4w

“(A) Appearance of *Sharpin*^{fl/fl}*K5*^{cre}*TCRα*^{-/-} mice.”

If you treat with a neutralizing TNFα antibody in vivo, can you prevent or treat disease?

As requested, we administrated 200 μg of an anti-TNFα Ab (clone XT3.11) or isotype control IgG (clone HRPN) intraperitoneally to 3-week-old *Sharpin*^{fl/fl}*K5*^{cre} mice. The mice received one injection every 3 days, and that was repeated eight times. We observed mild attenuation of skin disease after

blocking TNF α ; however, the inflammation was not completely suppressed, which indicates that TNF α is not solely responsible for disease progression (although it is critically involved in dermatitis in *Sharpin*^{fl/fl}*K5*^{cre} mice). This observation is not unexpected because cells with dysfunctional LUBAC are also susceptible to other death-inducing ligands, including FASL and TRAIL. We decided not to present these data in main or supplemental figures because involvement of TNF α in dermatitis in *Sharpin*^{fl/fl}*K5*^{cre} mice is not the main topic of the manuscript.

A

“(A) Administration of TNF α -neutralizing antibody on *Sharpin*^{fl/fl}*K5*^{cre}. Scale bar: 200 μ m.”

Lastly, out of pure interest – have you looked at the mucosa in any of these mouse models? There is a disease termed SJS/TEN (Stevens Johnson Syndrome/Toxic Epidermal Necrolysis) that is typically attributed to drugs but has some interesting potential overlaps in regards to t cell mediated keratinocyte death that might be a truer clinical corollary to your findings rather than generic dermatitis.

We appreciate this valuable comment. We have not been able to detect tissue disruption or abnormal inflammation in the intestine. However, as you suggest, investigating other mucosal tissues may be interesting and meaningful for showing the relevance of our study to SJS/TEN.

Reviewer #2

In this manuscript, authors present a series of studies suggesting that Sharpin deficiency in Tregs causes a cpdm-like skin inflammation, which is t cell-dependent dermatitis. Recovered expression of Sharpin in t lineages rescues cpdm mice from skin inflammation. Skin-oriented activated T cells promote apoptotic and necroptotic keratinocyte death and induce skin autoinflammation. Further, authors found T cell-expressed TNF- α can induce cell death to regulate skin inflammation. These functional studies provide new insight into the function of Sharpin and LUBAC in the regulation of inflammation. However, the work is somewhat descriptive and could be strengthened by including more in depth mechanistic studies.

Specific comments:

1. How Sharpin regulates Treg function was not sufficiently investigated. The authors used Sharpin-deficient Jurkat cells to show a reduction in the stability of LUBAC and activation of NF- κ B (sFig. 2a). However, it is unclear whether and how Sharpin regulates TCR signaling in Treg cells. To confirm the conclusion that ablating Sharpin in Tregs causes skin inflammation through partial impairment of TCR-mediated signaling, it is important to analyze the stability of LUBAC and signaling (such as NF- κ B activation) in Sharpin-deficient Treg cells.

As the reviewer points out, we have not performed biochemical analyses using *Sharpin*^{-/-} T_{regs}. Therefore, we examined the amount of LUBAC components expressed by primary T_{regs} isolated from *Sharpin*^{fl/fl}*Foxp3*^{Cre} or control mice. We found that loss of Sharpin from T_{regs} decreased the amounts of the two other LUBAC components, HOIP and HOIL-1L, possibly by inducing instability of the LUBAC complex (see new Fig. 1a); indeed, this was the case in Sharpin-KO Jurkat cells (see new Fig. S1C). Reduction in the amount of LUBAC results in partial impairment of TCR-mediated NF- κ B signal activation (see new Fig. 1c). We observed that loss of Sharpin

markedly dampened antigen-dependent IL-2 secretion (to the levels seen in HOIP deficiency) despite marginal attenuation of NF- κ B activation in T cell lines (see new Fig. S1E). Thus, the new results imply that loss of Sharpin impairs T_{reg} differentiation into effector T_{reg}, effector function, proliferation, and homeostasis by attenuating TCR-mediated signaling.

2. Fig.1h-K showed *Sharpin^{fl/fl}Foxp3^{Cre}* mice had lymphadenopathy with increased numbers of CD3⁺ T cells, B220⁺ B cells, and CD11b⁺ myeloid cells in pLNs. How about the immune cell infiltration in the skin in Treg-specific Sharpin-deficient mice.

Accumulation of CD3⁺T and CD11b⁺ myeloid cells in the skin of 10-week-old *Sharpin^{fl/fl}Foxp3^{Cre}* mice is shown in the new Fig. 6 (l, m) and the new Fig. 4b, respectively. Accumulation of B cells was not detected by immunofluorescence staining or flow cytometry analysis. As outlined below (A), we also analyzed younger mice (around 4 weeks old) prior to overt disease onset and found that they had mild skin inflammation, in which T cells and CD11b⁺ myeloid cells, but not B cells, had already infiltrated the epidermal tissues.

A

“(A) Percentage of the indicated epidermal infiltrates in the skin of young *Sharpin^{fl/fl}Foxp3^{Cre}*. Small circles indicate individual mice. Small horizontal lines indicate the mean (\pm s.e.m.). ns, $p > 0.05$; * $p < 0.05$ (two-tailed

Mann-Whitney U-test).”

3. Fig.2 and 3 showed skin inflammation observed in Treg-specific Sharpin-deficient mice is dependent on T cells. To rule out the role of other immune cells, the authors should deplete B220+ B cells and CD11b+ myeloid cells in Sharpin fl/flFoxp3Cre mice to analyze the skin inflammation.

To deplete B cells or myeloid cells, 4-week-old Sharpin^{fl/fl}Foxp3^{Cre} mice received an intraperitoneal injection of both anti-CD19 (1D3) and anti-B220 (RA3.3A1/6.1), or anti-Gr1 (RB6-8C5), or Isotype control (LTF2) antibodies; these injections were given five times per week. We did not observe any changes in progression of skin disease. We decided not to include the results in the revised manuscript because these are not related to the main aims of our manuscript.

A

“(A) Administration of antibodies to achieve B or myeloid cell depletion from Sharpin^{fl/fl}Foxp3^{Cre} mice. Scale bar: 200 μm.”

4. SFig.1 showed that keratinocyte-specific Sharpin-deficient mice developed cpdm-like dermatitis. Does Sharpin regulate apoptotic and necroptotic cell death in keratinocyte cells? Keratinocyte-specific Sharpin-deficient cells showed a reduction

in Foxp3 expression and an activated phenotype. What is the potential mechanism by which Sharpin-deficient keratinocytes regulate Treg phenotype?

Primary keratinocytes from *cpdm* mice are prone to apoptosis and necroptosis in culture (Kumari et al. eLife 2014). Because primary cultures do not take into account the effect of other cells, including T cells, we believe that deletion of Sharpin sensitizes keratinocytes to cell death. Also, we have shown that keratinocytes from *cpdm* mice harbor low amounts of other LUBAC components (Tamiya et al. J. Immunol. 2014).

Foxp3 expression by T_{regs} from *Sharpin^{fl/fl}K5^{Cre}* mice fell as dermatitis progressed, strongly indicating that skin inflammation affects T_{reg} phenotypes, possibly by inducing a cytokine storm (see new Fig. S4B, E–G). Accumulating evidence suggests that stimulation with inflammatory cytokines, including TNF α or IL-6/TGF β , triggers downregulation of Foxp3 expression by T_{regs} (Gao et al. PNAS, 2015, 112(25):E3246-54). Despite these findings, however, the underlying molecular mechanism is poorly understood.

5. In Fig.2, Sharpin fl/flLCKCre mice showed the reduced incidence of dermatitis and no changes in the population and phenotype of peripheral T cell lineages, compared with the Sharpin fl/flFoxp3Cre. The data suggest sharpin may be required for activation of CD4 or CD8 T cells. The authors should analyze the activation of CD4 and CD8 T cells from Sharpin fl/flLCKCre and wildtype control mice.

To examine the role of Sharpin on TCR signaling in conventional T cells, we analyzed the activation status of T cells after TCR stimulation by anti-CD3/CD28 antibodies. During the early phase of stimulation, we observed that induction of Nur77 expression (which is induced in an NFAT-dependent manner) by CD4⁺ and CD8⁺T cells from *Sharpin^{fl/fl}Lck^{Cre}*

mice was comparable with that by cells from control mice (see new Fig. 2f). However, activation of the NF- κ B signaling pathway was partially impaired in CD4⁺ and CD8⁺T cells from *Sharpin^{fl/fl}Lck^{Cre}* (see new Fig. 2e). Furthermore, we found defective differentiation of CD4⁺ and CD8⁺T cells into an activated state (CD25⁺ and/or CD69⁺) during the late phase (24–48 h after stimulation) (see new Fig. 2g). Collectively, these new results indicate that loss of Sharpin attenuates TCR-mediated activation of CD4⁺ and CD8⁺T cells.

6. Fig4d Sharpinfl/flK5Cre mice showed the tendency of reduced self-reactive CD4+ CD69+ T cells, The authors should repeat it by quantifying both the % and absolute cell number and by using more mice. It will also be interesting to perform parallel analysis using the Sharpinfl/flFoxp3Cre mice.

As requested, we have increased the number of mice analyzed by 2-fold; the results are shown in the new Fig. 4F. Despite the increased number of mice analyzed, we could not confirm reduced activation of T cells in *Sharpin^{fl/fl}K5^{Cre}* mice. Thus, the data from *Sharpin^{fl/fl}K5^{Cre}* and *Sharpin^{fl/fl}Foxp3^{Cre}* mice are presented in new Fig. 4F.

7. Fig.5 showed that apoptotic and necroptotic keratinocyte death were increased in both Sharpinfl/flFoxp3Cre and Sharpinfl/flK5Cre mice. What is the mechanism by which keratinocytes cell death induce skin autoinflammation and recruit CD11b+ myeloid cell infiltration?

As stated by the reviewer, it would be of great interest to identify the molecular mechanism underlying induction of skin autoinflammation upon keratinocyte cell death. Several previous reports indicate that keratinocyte death induces skin autoinflammation in *cpdm* mice; however, the underlying molecular mechanism is unclear. In this work, we utilized skin inflammation as an indicator to analyze the roles of LUBAC in T_{reg} cells and T

cell-mediated inflammatory changes. Thus, we believe that dissecting the underlying molecular mechanism is beyond the scope of this manuscript. However, we would like to discuss our idea regarding the mechanism here. We found that necroptosis is involved in the skin inflammation observed in *Sharpin^{fl/fl}Foxp3^{Cre}* mice because ablation of RIPK3 ameliorated dermatitis. In general, damage-induced necrosis, and a specific type of programmed cell death called necroptosis, release damage-associated molecular patterns (DAMPs), which are endogenous but locally segregated cytosolic or nuclear proteins; these induce innate immune responses by engaging with specific receptors on immune and non-immune cells. We believe that apoptosis drives skin inflammation in *Sharpin^{fl/fl}Foxp3^{Cre}* mice as well. An excess of apoptotic cells may abrogate appropriate clearance of apoptotic cells by phagocytes, resulting in release of DAMPs. Activated innate immune responses, which seem to be triggered by necroptosis, and possibly by apoptosis of keratinocytes, may induce production of inflammatory cytokines that contribute to further cell death, and chemokines such as CCL1 that recruit myeloid immune cells as shown below (A). This may generate the skin autoinflammation observed in *Sharpin^{fl/fl}Foxp3^{Cre}* mice.

A

“(A) Quantitative analysis of skin CCL1 mRNA expression. Small horizontal lines indicate the mean (\pm s.e.m.). *** $p < 0.001$ (two-tailed Mann-Whitney U-test). ”

8. Fig.6i showed activated T cells increased the expression of TNF-a, which can induce cell death. To study the increased cell death and skin inflammation, the authors should analyze the TNF- α expression in T cells from *Sharpin^{fl/fl}Foxp3^{Cre}* mice.

We appreciate this important comment. Because *Sharpin^{fl/fl}Foxp3^{Cre}* mice harbor a type-II inflammatory environment, including increased expression of IL-4, IL-5, and IL-10 (see new Fig. 7d), and serum IgG1 and IgE (see new Fig. S5l), T cells isolated from most tissues show suppressed expression of TNF α , as shown below. By contrast, skin-infiltrating T cells selectively acquired expression of TNF α on their surface (see new Fig. 6n); this supports our concept that skin-oriented T cells acquire expression of TNF α after TCR-mediated activation and then go on to induce keratinocyte cell death.

“(A) Percentage of TNF α -expressing T cells in lymphoid and non-lymphoid tissues.”

9. Fig.6j showed that the skin autoinflammation was attenuated in *Sharpin^{fl/fl}Foxp3^{Cre} TNF α ^{-/-}* mice and *Sharpin^{fl/fl}Foxp3^{Cre} RIPK3^{-/-}* mice. The authors should analyze the apoptosis and necroptosis of keratinocytes and dermal fibroblasts, showing that cell death induced skin autoinflammation.

As requested, we performed immunofluorescence staining to detect active caspase 3⁺ or pMLKL⁺ cells in the epidermis of *Sharpin^{fl/fl}Foxp3^{Cre}TNF α ^{-/-}* and *Sharpin^{fl/fl}Foxp3^{Cre}Ripk3^{-/-}* mice. The data are shown in the new Fig. 6k. Neither active caspase3⁺ nor pMLKL⁺ cells were detected in *Sharpin^{fl/fl}Foxp3^{Cre}TNF α ^{-/-}* mice, and no pMLKL⁺ cells were found in *Sharpin^{fl/fl}Foxp3^{Cre}Ripk3^{-/-}* mice.

Reviewer #3

The manuscript by Sasaki and colleagues studied the importance of LUBAC in Treg cells. The authors generated various conditional knockout mouse lines in which Sharpin or Hoip is specifically deleted in Tregs, T cells or Keratinocytes. The authors showed that knockout Sharpin in Tregs induced some similar skin phenotypes as those observed in global KO and keratinocyte specific KO of Sharpin, suggesting an important role of Tregs in the development of dermatitis. Mechanistically, reduced Treg function resulted in T cell mediated keratinocyte death through mainly the TNF- α pathway. The studies were generally well performed and data were comprehensive. Several issues however need to be further addressed.

Major points:

1) Some the terminologies were used in a very confusing way throughout the study. For example, autoimmunity and autoinflammation are very difficult to distinguish and probably overstated in the study, since the authors were never provided any concrete data to support any of the phenotypes from various mouse lines fitting the restrictive definitions of these two terminologies, such as autoantigen/autoreactive T cells, or overexpression of either IL-1 or TNF α from innate immune cells. Therefore, the authors should remove (or minimize) the use them especially in a comparative and contrast way throughout the paper.

We agree that the data from several mouse lines might not be sufficient to definitively distinguish autoimmunity from autoinflammation, although some manifestations of the diseases are compared in the new Fig. 4. However, without using this terminology, it is almost impossible to describe etiological differences between *Sharpin^{fl/fl}Foxp3^{Cre}* and *Sharpin^{fl/fl}K5^{Cre}* mice; the skin inflammation is similar, but etilogically is quite distinct. The former is mediated by T cells, whereas the latter is mediated by the innate immune system after activation by keratinocyte death. Thus, we have tried to minimize the use of this particular terminology throughout; however, we

cannot avoid using the words “autoimmunity” and “autoinflammation” completely.

2] Although the authors generated many different mouse lines, the paper was a little descriptive. It would be helpful, if the authors could provide some mechanistic insights on how Sharpin/LUBAC impacted the Treg function and development.

We thank the reviewer for the critical comment. As described in the response to the reviewer #2 (comment 1), we found that Sharpin deficiency in T_{regs} results in a reduction in the other two LUBAC components (see new Fig. 1a); this was also observed in Sharpin-KO Jurkat cells (see new Fig. S1C) and *cpdm* T cells (see new Fig. S3B). We performed intracellular FACS staining to monitor TCR-mediated signaling in T_{regs}. A small but significant impairment of NF- κ B signaling, but not ERK signaling, was observed in *Sharpin*^{-/-} T_{regs} upon antibody-mediated TCR stimulation, which indicates that loss of Sharpin downregulates TCR-mediated NF- κ B activation in T_{regs} by reducing the amount of LUBAC.

Minor points

3] In Figure 6, the authors demonstrated an important role of TNF α pathway in T cell mediated keratinocyte death. But there is no data distinguished the role of soluble or membrane bound TNF. So in line 311, the authors should not use “membrane-bound TNF α ”.

As requested, we removed the sentence from the “Results” section. However, since we confirmed increased expression of TNF α by skin-infiltrating T cells (see new Fig. 6p), we describe the possible role of membrane-bound TNF α in T cell cytotoxicity in the “Discussion” section.

4] The authors need to further edit the language throughout the paper.

We have asked a native English speaker to check the revised manuscript.

REVIEWERS' COMMENTS:

Reviewer #2 (Remarks to the Author):

The authors have adequately addressed my original concerns, and the revised manuscript has been substantially improved.

Reviewer #3 (Remarks to the Author):

The authors have sufficiently addressed my previous concerns.

Reviewer #4 (Remarks to the Author):

The authors have responded carefully and thoughtfully to a long list of critiques and suggestions by the reviewer. In many cases, additional experiments were done and results included in the manuscript. In other cases figures were reorganized, text was simplified, and other revisions were made. The revised manuscript reads very clearly, and I believe they have satisfactorily addressed all of the reviewers comments.

Minor criticism: CD69 expression may not represent activation in this case, but rather identify of the T cells as having a tissue resident phenotype.

Point-by-point response to reviewers' comments:

Reviewer #2 (Remarks to the Author):

The authors have adequately addressed my original concerns, and the revised manuscript has been substantially improved.

Reviewer #3 (Remarks to the Author):

The authors have sufficiently addressed my previous concerns.

Reviewer #4 (Remarks to the Author):

The authors have responded carefully and thoughtfully to a long list of critiques and suggestions by the reviewer. In many cases, additional experiments were done and results included in the manuscript. In other cases figures were reorganized, text was simplified, and other revisions were made. The revised manuscript reads very clearly, and I believe they have satisfactorily addressed all of the reviewers comments.

Minor criticism: CD69 expression may not represent activation in this case, but rather identify of the T cells as having a tissue resident phenotype.

We thank all of the reviewers for highly evaluating our revised manuscript. Regarding the minor criticism from reviewer #4, we agree that CD69 expressed on the T cells at local tissues is not an absolute maker for T cell activation. Thus, we have to admit that we could not completely demonstrate that local antigen recognition by skin resident T cells is required for this inflammatory cell death. However, the T cells accumulated into the skin might be pre-activated in lymphoid tissues. It is because that we found that activation of T cells (CD69⁺) in lymphoid organs by breakout of immune tolerance would promote migration into several local tissues to become resident-memory T cells, and that it has been shown that CD69⁺ resident-memory T cells rarely recirculate and lodge in lymphoid tissues after getting resident phenotypes.